# Machine learning uncovers cell identity regulator by histone code

Bo Xia[1,2,3,4,7], Dongyu Zhao[1,2,3,4,7], Guangyu Wang[1,2,3,4,7], Min Zhang[2,3,4], Jie Lv[1,2,3,4], Alin S. Tomoiaga[5], Yanqiang Li [1,2,3,4], Xin Wang [1,2,3,4], Shu Meng[2,3,4], John P. Cooke[2,3,4], Qi Cao [6✉], Lili Zhang[2,3,4✉] & Kaifu Chen [1,2,3,4✉]

Conversion between cell types, e.g., by induced expression of master transcription factors, holds great promise for cellular therapy. Our ability to manipulate cell identity is constrained by incomplete information on cell identity genes (CIGs) and their expression regulation. Here, we develop CEFCIG, an artificial intelligent framework to uncover CIGs and further define their master regulators. On the basis of machine learning, CEFCIG reveals unique histone codes for transcriptional regulation of reported CIGs, and utilizes these codes to predict CIGs and their master regulators with high accuracy. Applying CEFCIG to 1,005 epigenetic profiles, our analysis uncovers the landscape of regulation network for identity genes in individual cell or tissue types. Together, this work provides insights into cell identity regulation, and delivers a powerful technique to facilitate regenerative medicine.

[1] Center for Bioinformatics and Computational Biology, Houston Methodist Research Institute, Houston, TX, USA. [2] Center for Cardiovascular Regeneration, Department of Cardiovascular Sciences, Houston Methodist Research Institute, Houston, TX, USA. [3] Department of Cardiothoracic Surgeries, Weill Cornell Medical College, Cornell University, New York, NY, USA. [4] Institute for Academic Medicine, Houston Methodist Research Institute, Houston, TX, USA. [5] Business Analytics, CIS & Law Department, The O'Malley School of Business Accounting, Manhattan College, Riverdale, NY, USA. [6] Department of Urology, Robert H. Lurie Comprehensive Cancer Center, Chicago, IL, USA. [7] These authors contributed equally: Bo Xia, Dongyu Zhao, Guangyu Wang. ✉email: qi.cao@northwestern.edu; lzhang3@houstonmethodist.org; kchen2@houstonmethodist.org

A cell type is determined by the combinatorial function of its cell identity genes (CIGs), which jointly dictate phenotypes of the cell and play critical roles in differentiation, development, and disease. CIGs form an intricate regulation network, in which a set of master transcription factors governs the expression program of CIGs[1]. Transitions between a few cell types have been achieved in vitro by induced expression of master transcription factors[2] and hold great promise for cellular therapy[3]. However, constrained by limited ability to uncover the accurate catalog of CIGs, our understanding of cell identity is incomplete. Conventional methods define cell identity by several marker genes, a methodology that is widely recognized to be suboptimal. Without comprehensive information about cell identity fidelity, clinical application of progenitor-derived cells might be encumbered by phenotypic aberration or instability[4–9]. Thus, it is of pivotal importance to uncover the complete catalog of CIGs, which will provide a blueprint to facilitate basic investigation on cell identity, and to guide clinical application of engineered cells.

Scientists have been defining CIGs based on genome-wide expression profile[10], which is not yet an optimal method. Many genes expressed in a cell type are not related to cell identity. Alternatively, scientists also define CIGs based on the expression specificity analysis[10], which requires comparison between a query cell type and most, if not all, other cell types. It also requires distinguishing between cell-type-specific and biological-condition-specific genes, e.g., heat shock genes. Therefore, it is not cost-effective, if not impossible, to collect data for all cell types under all biological conditions for the expression specificity analysis. Further, the relationship between CIGs and cell-type-specific genes can be complicated and uncertain. Some identity genes may be highly cell-type-specific, whereas other identity genes may be expressed in multiple related cell types and have less expression specificity. As such, to define CIGs solely by their expression profile may be not optimal. It becomes apparent recently that CIGs are different from other expressed genes in the epigenetic mechanism to regulate their transcription. For examples, super enhancers and a unique broad pattern of H3K4me3 modification were found to regulate CIGs, whereas it is typical enhancer and sharp H3K4me3 modification that regulate other expressed genes such as housekeeping genes[11–14]. These discoveries suggest strong potential to systematically uncover CIGs on the basis of epigenetic signature analysis.

Here we develop Computational Epigenetic Framework for Cell Identity Gene (CEFCIG), a computational framework to uncover CIGs and further define their master regulators. On the basis of machine learning, CEFCIG reveals unique histone codes for transcriptional regulation of reported CIGs and utilizes these codes to predict CIGs and their master regulators with high accuracy. Applying CEFCIG to 1005 epigenetic profiles, our analysis uncovers the landscape of regulation network for identity genes in individual cell or tissue types. Together, this work provides insights into cell identity regulation and delivers a powerful technique to facilitate regenerative medicine.

## Results

**Prediction of CIGs by CIGdiscover on the basis of the unique histone codes.** To build a solid foundation for study on CIGs, we performed a thorough review of literature (see "Methods") to develop Cell Identity Gene Data Base (CIGDB), the first database for manually curated known CIGs. Our meticulous review of literature motivated the use to define CIGs as belonging to at least one of four categories: (1) master transcription factors, which drive the differentiation towards a cell type when their expression is ectopically induced in another cell type; (2) required

transcription factors, whose depletion impaired the differentiation towards a specific cell type; (3) genes required for key functions or phenotypes of a cell type; (4) genes that were widely used as markers for a cell type. Together, we curated 247 known identity genes for ten well-defined cell types (Supplementary Data 1). These genes include 42–82 identity genes in each of the four categories (Fig. 1a) and 18 to 36 identity genes for each of the ten cell types (Fig. 1b). As was reported before, chromatin immunoprecipitation sequencing (ChIP-seq) signal of activating epigenetic marks, such as H3K4me3, H3K4me1, and H3K27ac, showed a unique broad pattern of enrichment at individual known CIGs (Supplementary Fig. 1A, B), e.g., SOX2 in H1 human embryonic stem cell (hESC)[2], GATA2 in endothelial cells (ECs)[15], LMO2 in CD34 + hematopoietic stem cells (HPCs)[16], and PAX6 in mid-radical glial cells (MRGs)[17] (Supplementary Fig. 1C). These known CIGs have a high but widely distributed Tau index of expression specificity, with their index values ranging from ~1 to 0.6 and ranked from top to median among all genes (Supplementary Fig. 1d).

We noted that different histone modifications differ in distribution pattern on chromatin and thus require different parameters to analyze their ChIP-Seq data. We therefore developed GridGO, a grid-based genetic algorithm to automatically optimize the detection of histone modification features for CIGs (Supplementary Fig. 2a). The algorithm aggressively searches for optimal cutoff for each of three important parameters in ChIP-Seq analysis, including the height cutoff for peak calling and the upstream or downstream distance cutoffs for assigning a peak to a nearby Transcription Start Site (TSS). To investigate additional histone modification features for CIGs, we further developed algorithms in GridGO to analyze six statistical features for each ChIP-Seq enrichment peak. These features include peak width, height, skewness, kurtosis, total signal, and coverage in a given genomic region (Supplementary Fig. 2b). GridGO effectively improved $P$-values of differences in most of histone modification features between CIGs and the remaining genes (Supplementary Fig. 2c, d). We further compared the receiver operating characteristic (ROC) curve for recapturing the curated CIGs using each feature before and after GridGO optimization. Sixteen signatures, each defined as one feature of one histone modification, display significant improvement (Supplementary Fig. 2e-g). Intriguingly, H3K4me3, H3K4me1, and H3K27ac showed significant difference in each of their features between known CIGs and random control genes (Fig. 1c). However, H3K27me3, one of the most studied repressive histone modifications, displayed little such difference (Fig. 1c).

By combining gene expression profile with histone modification signatures, we developed a logistic regression model, CIGdiscover, for CIGs discovery (Supplementary Fig. 3a). A histone modification may have distinct biological implications when it appears on the promoter vs. on the gene body. Therefore, for each histone modification feature, CIGdiscover calculated a value for the associated promoter and further a value for the associated gene body. These together led to 49 signatures. Considering that some peak features occasionally show considerable correlation with each other, CIGdiscover tried the backward feature elimination and forward feature construction methods to select an optimal combination of features for prediction of CIGs. Intriguingly, the backward method did not improve the model, whereas the forward method successfully removed 40 of the 49 signatures to achieve an optimal prediction accuracy (Supplementary Fig. 3b). The most frequently selected features are the peak width and kurtosis, of which both were selected for the three histone modifications associated with gene activation (Fig. 1d). H3K4me3 peak width on the gene body and H3K4me1 peak width on the promoter are the most useful signatures. No

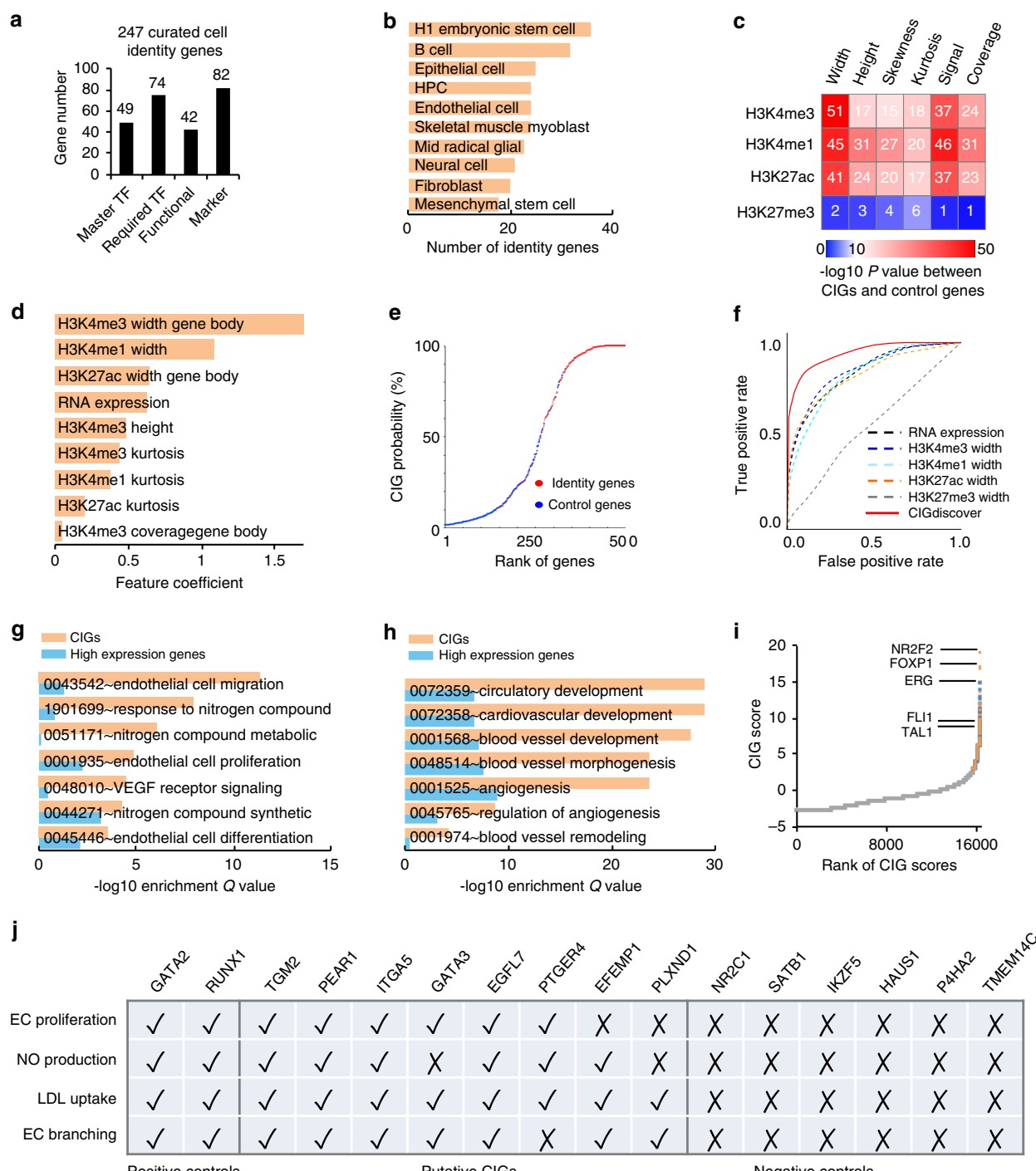

**Fig. 1 CIGdiscover uncovers cell identity genes (CIGs) on the basis of the unique histone codes for their transcriptional regulation. a**, **b** The number of curated CIGs for each category (**a**) or cell type (**b**). **c** Heatmap to show −log10 P-value for difference in histone modification feature between CIGs and random control genes. P-values determined by Wilcoxon test. **d** Bar plots to show importance, as determined by coefficient, for each individual feature used by the logistic regression model in CIGdiscover. **e** The probability calculated by CIGdiscover to be CIG plotted for each curated known CIG and control gene. **f** ROC curves to show accuracy for CIG prediction using CIGdiscover or other methods. **g**, **h** Barplot to show enrichment of endothelial (**g**) or cardiovascular (**h**) related pathways in CIGs uncovered by CIGdiscover or in high expression genes for HUVECs. **i** Endothelial CIG score determined by CIGdiscover for each gene. Predicted EC CIGs that were found to have reported endothelial functions are marked in orange color. Predicted EC CIGs that were not found to have reported endothelial functions are marked in blue color. Genes not predicted to be EC CIGs are in gray color. **j** Summary of experiment verification results for predicted CIGs, positive control genes, and negative control genes.

feature associated with H3K27me3 were selected by the model. Intriguingly, RNA expression value is useful in prioritizing negative control genes, but is less useful in prioritizing CIGs (Supplementary Fig. 3c). Therefore, the false positive rate for the

model that was based only on RNA expression level could be still low when the true positive rate reaches to a medium level of 50% (Fig. 1f). The CIGdiscover successufly recaptured known identity genes (Fig. 1e), with a sensitivity value 0.89, specificity value 0.92,

precision value 0.91, negative predictive value 0.90, and significantly outperformed each single signature that was empirically used by biologists (Fig. 1f).

As a proof of principle, we utilized CIGdiscover to define CIGs for human umbilical vein ECs (HUVECs). As expected, the CIGs are significantly enriched with endothelial (Fig. 1g) and cardiovascular functions (Fig. 1h). For comparison, we observed moderate enrichment of these pathways when we use the same number of top genes predicted by high expression to perform pathway analyses. Thorough literature review revealed that 255 (42.9%) of the predicted CIGs for EC have reported role in endothelial differentiation, phenotypes, or functions (Supplementary Data 2). Examples of known endothelial CIGs recaptured by our model include the vein endothelial transcription factors NR2F2[18], FOXP1[19], ERG[20], FLI1[15], and TAL1[21] (Fig. 1i). Similarly, CIGdiscover successfully recaptured CIGs reported for other cell types, e.g., the SOX2[2], KLF4[2], MYC[2], and NANOG[2] for mESC; the RUNX1[22], MEIS1[23], ETV6[24], MECOM[25], MYB[26], and LMO2[16] for CD34+ hematopoietic progenitor cell; and the FOXG1[27], PAX6[28], SOX1[29], SOX2[30], POU3F2[31], HES5[32], NOTCH1[30], and ASCL1[33] for glial cell (Supplementary Fig. 4a). As expected, the CIGs show strong and broad enrichment of the histone modifications associated with gene activation (Supplementary Fig. 4b); however, genes predicted using RNA expression level showed significantly higher expression level when compared with random control genes or CIGs predicted by the full model, whereas CIGs predicted by our approach are marked by the broader H3K4me1/3 and H3K27ac when compared with random control genes or genes predicted by RNA expression (Supplementary Fig S5–8). Further, the CIGs for the other nine cell types are enriched in their associated cell-type-specific functions as well (Supplementary Fig. 4c). Although CIGdiscover did not utilize information related to cell-type specificity, the histone modification signatures and RNA expression of the CIGs show strong specificity to their associated cell types (Supplementary Fig. 4d). These analyses indicated that CIGdiscover successfully recaptured known CIGs. We further experimentally verified endothelial CIGs that were not reported before. We randomly selected eight candidates to disrupt each of them in HUVECs by CRISPR-Cas9 system (Supplementary Fig. 10) and then test the effect on four important phenotypes of EC, i.e., proliferation, NO production, low-density lipoprotein (LDL) uptake, and EC tube formation. We also tested two positive control genes and six negative control genes that expressed in HUVECs (Supplementary Fig. 9). Together, all eight predicted CIGs were verified to have significant effects on at least two of the tested four phenotypes of ECs, whereas none of the negative control genes have significant effect on any of the four phenotypes (Fig. 1h and Supplementary Figs. S11–14).

**Performance of CIGdiscover on the small and noisy training datasets**. Notably, to uncover identity genes for a query cell type, CIGdiscover does not have to be trained with known identity genes of the same cell type. Both literature and our analysis (Supplementary Fig. 1a–c) indicated that the existence of unique epigenetic signatures (e.g., broad H3K4me3 and super enhancer) at CIGs but not at other genes (e.g., housekeeping genes) is a phenomenon across different cell types[12,14]. For instance, embryonic stem CIGs displayed broad H3K4me3 in ESCs and neural CIGs also displayed broad H3K4me3 in neural cells (Supplementary Fig. 1c). Therefore, when our machine-learning model learned that stem CIGs display features such as broad H3K4me3 in ESCs, it would be able to use the broad H3K4m3 in neural cells to define identity genes of neural cells. In addition, when we combine CIGs of ESCs and their H3K4me3 signatures

in ESC with CIGs of neural cells and their H3K4me3 signatures in neural cells, the machine-learning model will be able to learn that CIGs of ESCs and NPCs both display broad H3K4me3 in their associated cell types. Therefore, the model will be able to use broad H4K4me3 in a third cell type to predict CIGs for that third cell type. To verify this expectation, we performed a comparison between two variants of CIGdiscover. In one variant, the query cell types and the training cell types are the same (parallel test), whereas the query cell types are different from the training cell types in the other variant (cross test) (Supplementary Fig. 15a). The result indicated that there is little difference in performance between the two variants. We next performed down sampling to test how many training cell types are required to arrive at optimal performance. The result indicated that using up to three of the ten cell types has been good enough to achieve the best prediction accuracy (Fig. 2a, b). Therefore, the current number of training cell types has been large enough to arrive at optimal performance for CIGdiscover.

We further performed down sampling to test how many training genes are required to arrive at optimal performance for CIGdiscover and found that using up to 15% (37) of the 247 curated CIGs has allowed the model to perform as well as using all the curated CIGs (Fig. 2c, d). Considering that the number of non-identity genes in a cell might be much larger than the number of CIGs, we questioned whether this imbalance influences the performance of CIGdiscover. We thus analyzed how the accuracy of CIGdiscover changes in response to different number of non-identity genes that we used to train the model. Because the number of reported positive CIGs is small and thus it is hard to further manually curate additional positive CIGs from literature. We decided to gradually increase the number of non-identity genes from 1 to 20 folds more than that of CIGs. We found that the performance of CIGdiscover changed little in response to this increase of imbalance (Supplementary Fig. 15b). We next tested whether CIGdiscover could be resilient enough to a small proportion of potential false positives or false negatives when the training identity genes were collected from literature. We performed a simulation by swapping the labels of CIGs and non-identity genes. Even when 30% of the control genes were from CIGs and thus are false control genes, the AUC of ROC only reduced from 0.93 to 0.89 (Supplementary Fig. 15c). As expected, the performance of CIGdiscover was ruined when the noise ratio reached 50%, suggesting that the good performance at low noise ratio was not due to an overfitting effect. Intriguingly, the model is slightly more resilient to false positives (by labeling random control genes as CIGs) when compared with false negatives (by labeling CIGs as random control genes). We further tested whether different categories of CIGs that we defined from literature are associated with different histone modification signatures and thus need different prediction models. Although they represent different aspects of cell identities, we found that using one category of CIGs to train CIGdiscover was good enough to predict all categories of CIGs (Supplementary Fig. 15d). The performance also remained comparable when one category was used to train CIGdiscover for the prediction of another category (Fig. 2e), indicating that different categories of CIGs undergo similar epigenetic regulation mechanism.

We next tested whether the performance of CIGdiscover was effectively improved by combining the different histone modifications. The full model that combined all 49 signatures from the four different histone modifications and RNA expression significantly outperformed the simplified versions that only combines all 12 features of H3K27ac, but just slightly outperformed other two simplified versions that each used H3K4me3 or H3K4me1 alone (Fig. 2f). Increasing cell types or increasing training datasets did not further improve the accuracy

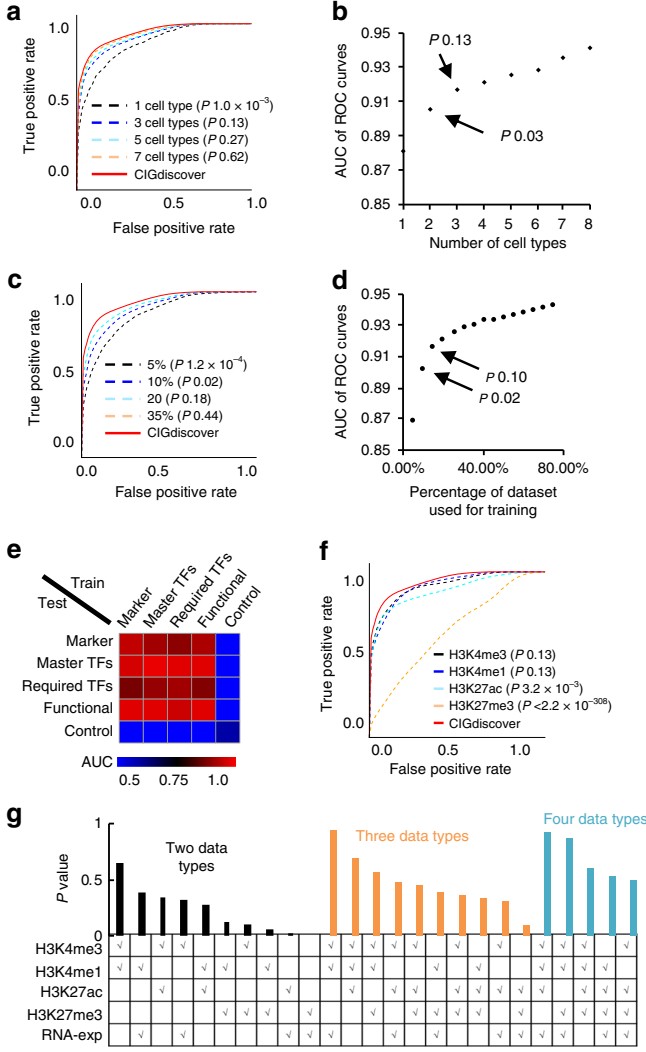

**Fig. 2 CIGdiscover is robust and thus resilient to small or noisy training dataset. a** ROC curves to show the performances of CIGdiscover and its variants trained based on data from different number of cell types. **b** Scatterplot to show the ROC AUC values for CIGdiscover variants trained by data from different number of cell types. **c** ROC curves to show performance of CIGdiscover and its variants trained with smaller number of known CIGs. **d** Scatterplot to show the AUC of ROC for CIGdiscover variants trained by smaller number of CIGs. **e** Heatmap to show AUC value of ROC for CIGdiscover variants trained by one category but tested by individual other categories of CIGs. **f** ROC curves to show performance of CIGdiscover and its variants that each only utilized all features of one type of histone modification. **g** Barplot to show P-values of difference in performance between CIGdiscover and its variants that each utilized a smaller combination of histone modifications and RNA expression. P-values labeled in **b** and **c** indicate difference between ROC curves for CIGdiscover and the associated variants.

of simplified versions, as the accuracy was saturated quickly (Supplementary Fig. 16). By combining information from multiple types of histone modifications or expression data, the model gained performance slightly (Fig. 2g). Therefore, CIGdiscover does not have to combine multiple types of data to achieve an optimal performance, and H3K4me3 and H3K4me1 appeared to be the most useful data types for CIGdiscover.

**Uncover master transcription factors in the regulation network of CIGs.** Intriguingly, CIGs defined by CIGdiscover are 6.7-fold

enriched with regulation relationship between each other, as indicated by significantly large number of network edges (Fig. 3a). Further, the CIGs are 2.4-fold enriched with transcription factors (Fig. 3b). As expected, when compared with the same number of randomly selected control transcription factors, known master transcription factors in the CIGs catalog have larger number of parental edges, in addition to a larger number of children edges, in their associated regulation network of identity genes (Fig. 3c), suggesting that master transcription factors serve as hubs in the network. These results motivated us to develop CIGnet, a network method to identify master transcription factors for the CIGs defined by CIGdiscover. Leveraging the regulation relationship defined between genes in the CellNet database[1], CIGnet is the first to utilize an unbiased logistic regression method to learn an optimal combination of network features for discovery of master transcription factors. In contrast to conventional empirical methods that often intuitively used the number and expression level of downstream target genes to score the importance of a regulator, our machine-learning methods revealed that the network features related to upstream regulators are more important for determining the master transcription factors (Fig. 3d). These features include the number, closeness, and CIG score of upstream regulators. This result suggests that many CIGs act in concert to regulate the master transcription factors, which in turn further regulate CIGs. CIGnet predicted master transcription factors of CIGs with a sensitivity value 0.88, specificity value 0.88, precision value 0.88, negative predictive value 0.89, and outperformed conventional methods that empirically use network, expression level, or differential expression to prioritize master transcription factors (Fig. 3e). Histone modification signatures and RNA expression of predicted master transcription factors manifested strong cell-type specificity (Fig. 3f). We experimentally tested eight putative master transcription factors for EC and further tested two positive and two negative control transcription factors that expressed in HUVECs (Supplementary Fig. 9). Using CRISPR-Cas9 system to disrupt each factor, we observed a significant reduction of differentiation from human pluripotent stem cells (PSCs) to ECs (VE-cadherin+, CD31+) for seven out of the eight tested putative master transcription factors and for both of the two positive control transcription factors, whereas none of the two negative control transcription factors show a significant effect (Fig. 3g).

**A comprehensive landscape of CIGs.** We finally wrapped CIGDB, GridGO, CIGdiscover, and CIGnet as a CEFCIG research (Fig. 4a) and applied it to epigenomic datasets (Supplementary Data 3) from 57 cell types, to investigate the landscape of identity genes (https://sites.google.com/view/cigdb) for each cell type. As expected, related cell types were clustered closely on the base of CIG scores of CIGs (Supplementary Fig. 17a). In addition, most of the uncovered CIGs appeared to be cell-type specific, in which 48% of CIGs belong to one to three cell types (Fig. 4b). The CIGs were enriched in functional pathways for their associated cell types, but not for the other cell types (Fig. 4c, d). Further gene regulation network analysis indicated that the CIGs formed an internal network within each specific cell type, marked by significantly larger number of edges and closer distance between nodes in internal network relative to external edges (Fig. 4e, f). In contrast, although high expression genes also show some specificity to associated cell types (Supplementary Fig. 17b), they failed to enrich or are much less enriched in the functional pathways for their associated cell types (Fig. 4d). Also, high expression genes failed to form an internal gene regulation network within their specific cell types (Supplementary Fig. 17C). These conclusions are reproducible in the landscape of tissue

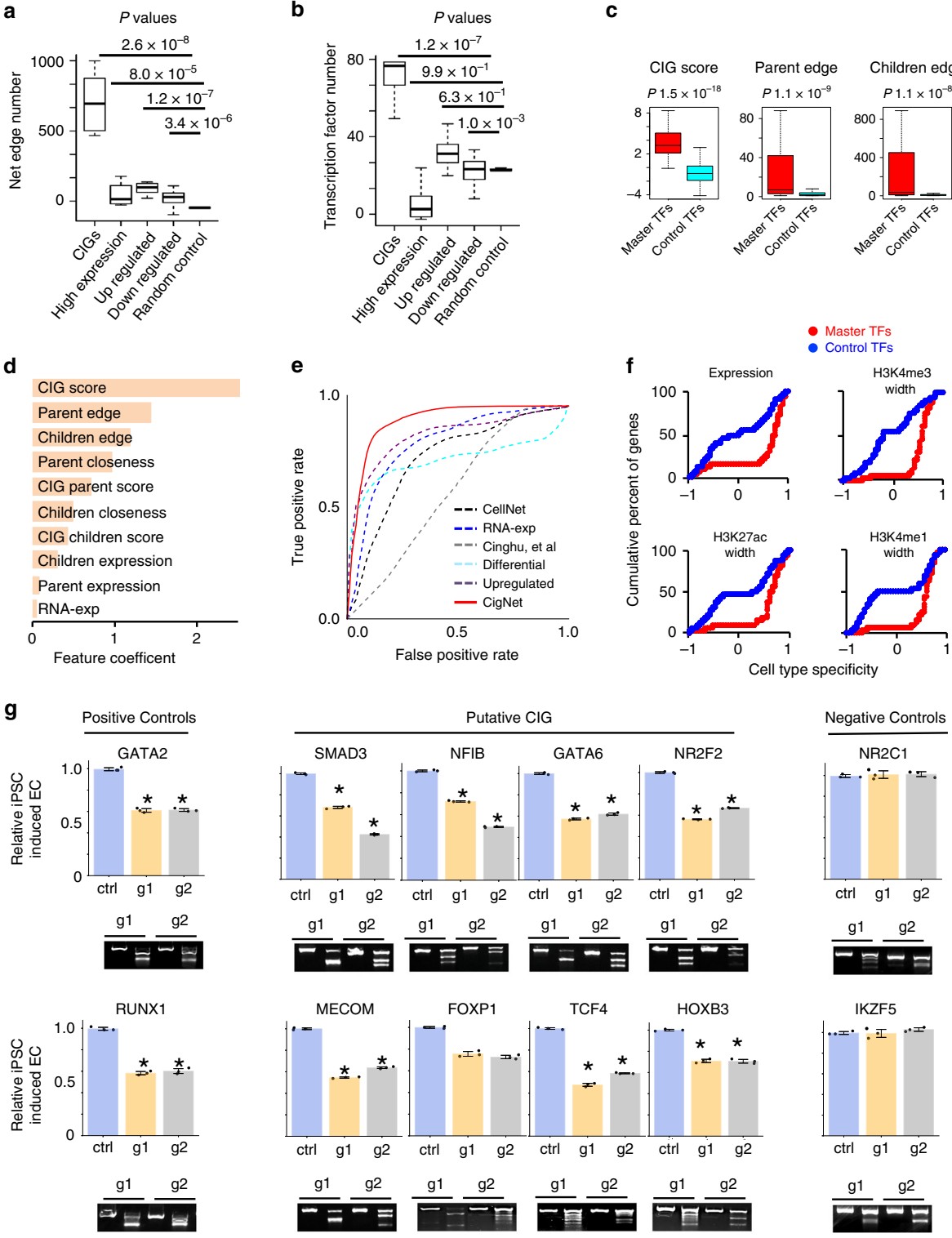

identity genes (Supplementary Fig. 17d–f). Our analysis further revealed that master transcription factors have their own unique features when compared with other transcription factors. The master transcription factors uncovered by CEFCIG showed significantly higher CIG score (Fig. 4g), as well as larger number of children edges and parental edges (Fig. 4h, i) when compared with control transcription factors. Notably, master transcription factors were connected to more transcription factors within the CIG catalog (Fig. 4j, k).

## Discussion

A major obstacle to the development of technology for cell fate conversion is the lack of accurate information for CIGs. Biologists often empirically define CIGs based on expression value or a single epigenetic signature[10,12,14], which is not optimal due to limited information provided a single signature. Recently, epigenetic signatures were used as the inputs of various computational models to define cell-type-specific regulatory relationship between transcription factors and their target genes on the basis

**Fig. 3 CIGnet as a network model to uncover master transcription factors in the regulation network of CIGs. a, b** Box plot to show average number of network edges between genes (**a**) and average number of transcription factors (**b**) within each gene group. For fair comparison, each gene group was defined to have the same number of genes. **c** Box plots to show CIG scores (left), number of parental edges (middle), and number of children edges (right) associated with reported master transcription factors and random control transcription factors. Box plots: center line is median, boxes show first and third quartiles, whiskers extend to the most extreme data points that are no more than 1.5-fold of the interquartile range from the box. *P*-values were determined by Wilcoxon's test. **d** Barplot to show coefficient for individual network features utilized by the regression model in CIGnet. **e** ROC curves to show performances of CIGnet and other conventional methods for recapturing known master transcription factors. **f** Cumulative percentage of genes plotted against cell-type specificity of gene features. **g** Barplot to show changes in percentage of induced endothelial cells derived from PSCs in which individual master transcription factors were disrupted using the CRISPR-Cas9 system relative to that from wild-type PSCs. Two gRNAs g1 and g2 were tested for each gene. T7 endonuclease cleavage assay is used to confirm cutting efficiency of the CRISPR-Cas9 system. *P*-values determined by Student's *T*-test. *$P < 0.05$. Source data are provided as a Source Data file.

of three-dimensional chromatin interactions[34,35], expression profiles[36], and binding of transcription factors[37]. When combined with cell-type-specific genes, these methods have the potential to be further utilized to identify important transcription factors for the associated cell type. Although some CIGs can be very specific to one cell type, some other CIGs might be expressed in multiple related cell types and thus have different degrees of expression specificity. Therefore, CIGs are not conceptually the same as cell-type-specific genes. Here we developed the machine-learning model CIGdiscover that quantitatively determines the optimal combination of epigenetic signatures to define CIGs and further developed the network model CIGnet to identify the master transcription factors that regulate CIGs. These methods address the most pressing constraints and fundamental aspects of technologies for cell identity conversion and will thus help remove critical barriers impeding the development of cell therapy.

We developed a bioinformatics toolkit for ChIP-Seq analysis of epigenetic features associated with CIGs. We previously developed DANPOS and DANPOS2[11], which are among the earliest tools for analysis of genome-wide chromatin marks such as nucleosome positioning, histone modification, the binding of chromatin protein, or chromatin openness. Leveraging these successes, we developed a next-generation toolkit GridGO in this study. Distinct from existing tools that simply use statistical significance to define epigenetic feature, GridGO is a bioinformatics technique that utilize biological significance for ChIP-Seq detection of epigenetic feature. GridGO further differentiate from existing tools by automatic customization of ChIP-seq bioinformatics pipeline for each unique research goal and thus provide a solution to handle heterogeneity of chromatin marks in bioinformatics analysis. Different research goals often require different parameter values to achieve optimal performance of a bioinformatic tool. It requires extensive user experience to optimize the parameters and is one of the most labor-intensive parts of bioinformatics analysis. Our GridGO algorithm represents an innovation to address this challenge by automatically optimize parameters on the base of a grid-based parameter optimization method. Finally, GridGO integrate six features of enrichment peaks, i.e., height, width, total signal, coverage, skewness, and kurtosis, for a given type of chromatin mark at each gene. These strategies enabled us to unravel epigenetic signatures of CIGs and thus to gain insight into epigenetic mechanisms for transcriptional regulation of CIGs.

We developed a manually curated database of reported CIGs and further developed two machine-learning models, CIGdiscover and CIGnet, for prediction of CIGs. There are thousands of publications directly using the terminology "cell identity gene." Strong interests in CIGs are particularly enhanced by a recent discovery of unique mechanisms, e.g., super enhancer, to regulate the expression of these genes. Despite this broad and increasingly strong interest, we found no literature providing a clear definition or catalog of CIGs. We thus compiled the standard for defining

CIGs based on a thorough literature review and delivered a manually curated catalog of known CIGs. The CIGdiscover is a bioinformatics tool specifically for CIGs discovery. Our CIGdiscover algorithm successfully outperformed each single epigenetic signature empirically used by biologists in uncovering CIGs. CIGs from different cell types shared the same types of epigenetic signatures. Therefore, a model trained by a limited number of cell types are good enough to predict CIGs for many other cell types. This makes CIGdiscover useful when many new cell types are identified in future, e.g., by single-cell sequencing technology. Although the synergistic effect of combining different types of epigenomic data improved the performance, combining two types associated with gene actiavtion can be good enough to maximize predition accuracy. This feature of CIGdiscover makes it applicable towards cell types with limited types of epigenomic datasets. To convert the regulation network of CIGs into an importance score to uncover master transcription factors, we developed CIGnet, an algorithm that integrates prior bio-phenotypic knowledge to automatically learn a set of optimal parameters for the conversion. These network parameters have important implications for understanding the transcriptional regulation of identity genes during cell-type transition. On the basis of these innovations, CIGnet predicts master transcription factors on the basis of calculated importance of individual network features.

Taken together, our study provided important insights into epigenetic regulation of CIGs and cell-type determination, and delivered innovations that are of broad interests to a wide spectrum of research communities. Our bioinformatics toolkit is designed for prediction of CIGs using histone modification data; however, the framework is amenable to incorporation of additional types of epigenomic data, e.g., chromatin-binding transcription factors, chromatin accessibility profiling data generated by DNase-Seq, or DNA methylation data generated using MRE-Seq, and could be applied to study other categories of genes. In addition, we developed CIGDB, the database for researchers to easily access the comprehensive landscape of CIGs that have been manually curated or systematically uncovered on the basis of >1,000 epigenetic profiles, as well as to easily deposit newly discovered CIGs. These tools together addressed the most pressing constraints on and fundamental aspects of cell identity research and thus will help overcome critical barriers impeding the development of cellular therapy. This work further provided important insights into epigenetic code for transcriptional regulation of CIGs and revealed unique mechanisms for network regulation of cell identity by master transcription factors.

## Methods

**Curating CIGs.** We first performed an intensive query of PubMed using the search expression "cell identity"[Title/abstract] OR "cell marker"[Title/abstract], which returned 7581 PubMed abstracts. We then searched the abstract for names of 297 cell types listed in the SHOGoiN database and ranked cell types by number of associated abstracts. To retrieve CIGs, we then conducted a manual literature review for the ten top-ranked cell types that also have RNA-sequencing (RNA-seq)

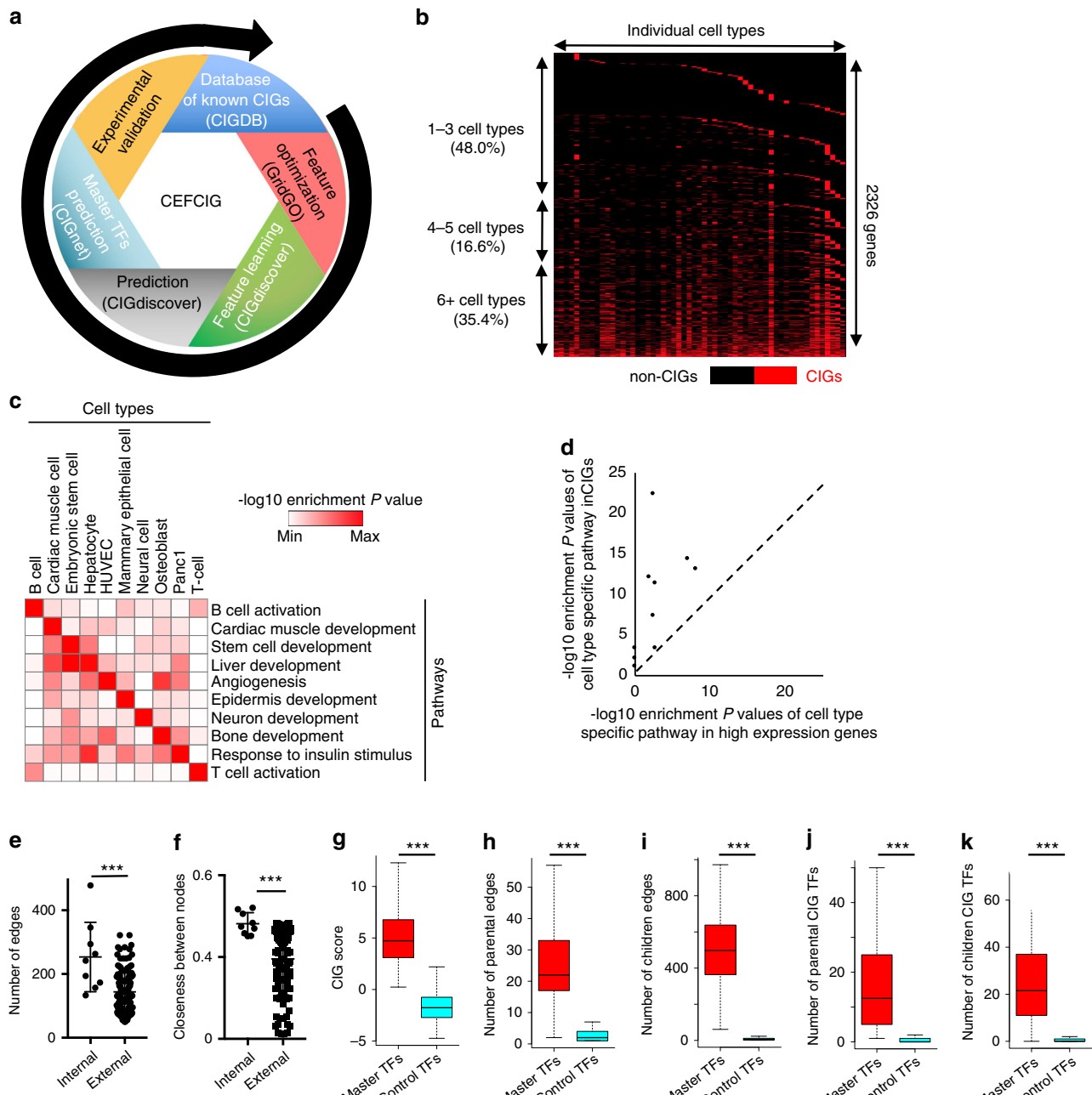

**Fig. 4 A comprehensive landscape of CIGs uncovered by CEFCIG. a** A cartoon to show CEFCIG framework to uncover cell identity genes and their master transcription factors. **b** Heatmap to show the specificity of CIGs from 57 cell types. **c** Heatmap to show the enrichment of individual pathways in CIGs defined for individual cell types. **d** Scatterplot to show −log10 enrichment $P$-value of cell-type-specific pathway in identity genes and high expression genes for each representative cell type. Each dot indicates one cell type. **e**, **f** One-dimensional scatterplots for network edge number (**e**) and closeness score (**f**) between CIGs from the same cell type (internal) or from different cell types (external). **g–k** Box plots to show CIG score (**g**), number of parental edges (**h**), number of children edges (**i**), number of parent transcription factors (**j**), and number of children transcription factors (**k**) associated with individual transcription factor groups. $P$-values were determined by Wilcoxon's test (**g**–**k**). ***$P < 0.001$. Box plots: center line is median, boxes show first and third quartiles, whiskers extend to the most extreme data points that are no more than 1.5-fold of the interquartile range from the box.

data and ChIP-Seq data for the histone modifications of H3K4me3, H3K4me1, and H3K27me3. We also defined control genes by requiring that their names did not appear together with the name of the given cell type in literature or any of five major annotation database, i.e., the Entrez Gene, Gene Cards, Ensembl, Gene Ontology, and KEGG. We then further selected a random subset of the control genes, so that number of control genes in the subset is the same as number of our curated CIGs.

**Data collection**. The RNA-seq, H3K4me3, H3K4me1, H3K27ac, and H3K27me3 ChIP-seq data for the ten well-defined cell types (H1-hESC, CD34 + HPC, B cell, HUVECs, human mammary epithelial cells, neural cells, MRG cell, normal human

lung fibroblast, mesenchymal stem cell, human skeletal muscle myoblast) and landscape analysis are downloaded from GEO database and ENCODE project website (https://www.encodeproject.org/)[38].

**ChIP-seq and RNA-seq data analysis**. Human reference genome sequence version hg19 and UCSC Known Genes were downloaded from the UCSC Genome Browser website[39]. RNA-seq raw reads were mapped to the human genome version hg19 using TopHat version 2.1.1 with default parameter values. Expression value for each gene was determined by the function Cuffdiff in Cufflinks version 2.2.1 with default parameter values. Afterwards, quantile normalization of gene expression values was performed across cell and tissue types.

For ChIP-seq data, reads were first mapped to hg19 human genome by Bowtie version 1.1.0:

bowtie -p 8 -m 1 --chunkmbs 512 –best *hg19_reference_genome fastq_file*

Wig file is generated using DANPOS 2.2.3:

python danpos.py dpeak *sample* –b *input* --smooth_width 0 -c 25000000 --frsz 200 --extend 200 –o *output_dir*

Quantile normalization is performed using DANPOS 2.2.3:

python danpos.py wiq --buffer_size 50 *hg19.chrom.sizes.xls wig* –reference *reference.qnor.sort.wiq* --rformat wiq --rsorted 1

By this method, ChIP-seq data from different cell and tissue types were all normalized to have the same quantiles.

Bigwig is generated using the tool WigToBigWig with the following command line:

*wigToBigWig -clip sample.bgsub.Fnor.wig hg19.sizes.xls sample.bw*

The tool WigToBigWig was downloaded from the ENCODE project website (https://www.encodeproject.org/software/wigtobigwig/)[38]. The "hg19.sizes.xls" in the command line is a file containing the length of each chromosome in the human genome. We then submitted the bigWig file to the UCSC Genome Browser (https://genome.ucsc.edu) to visualize ChIP-seq signal at each base pair[39,40].

Peak calling and feature value calculation are performed using the GridGO function in the CEFCIG framework (detailed algorithm is described below).

In feature value calculation, skewness and kurtosis values are centered on zero. If no signal is being detected in the peak calling region, skewness and kurtosis are set to zero.

**GridGO algorithm**. We developed GridGO, a grid-based genetic method to optimize bioinformatics parameters for detecting epigenetic signature of CIGs. We use GridGO to optimize three important parameters, including the height cutoff to define ChIP-seq enrichment peak, the upstream distance cutoff to assign a peak to a nearby gene, and the downstream distance cutoff to assign a peak to a nearby gene. However, GridGO is designed to also allow optimizing different number of parameters. For simplicity, we will describe details of the algorithm by an example in which the upstream and downstream distance cutoffs to assign a ChIP-seq peak to nearby genes are set to be the same, so that GridGO will optimize only two parameters including the height cutoff to define ChIP-Seq enrichment peak and the distance cutoff to assign a peak to a nearby gene (Supplementary Fig. 2a). In the first iteration of optimization, the entire two-dimensional parameter space will be divided into $m$ equal-size grids. Then the parameter values in the center of each grid will be used to define ChIP-Seq enrichment peaks and to assign the peaks to nearby genes. Afterwards, $P$-value of difference in a peak feature (epigenetic signature) between CIGs and control genes will be determined by Wilcoxon's test. The grid with the lowest $P$-value will be the optimal grid saved for the second iteration. In the second iteration, the grid saved in the first iteration will be divided into a new set of $m$ small grids, which will be tested as the previous iteration to select an optimal grid saved for the third iteration. Such genetic evolution of parameter grid keeps going until the number of iteration become larger than a given value $n$ or the new optimal grid is not better than the previous optimal grid. To estimate a potential overfitting effect, we used only 80% of training genes in the GridGo optimization and build the CIGdiscover model based on parameters optimized by these genes. Then the performance of CIGdiscover on these 80% genes and the rest 20% genes were compared, and little overfitting effect was observed.

**Backward and forward feature selections**. In backward feature elimination, all features are included in the model at the beginning. In each round of iteration, after trying to remove individual features from the model and test the influence on the model, one feature with least impairment to the performance of the model is removed. In contrast, in forward feature construction, there is no feature in the model in the beginning. In each round of iteration, after trying to add individual features from the feature pool and test the influence on the model, the feature that led to the best improvement to the model was added into the model. The performance is measured by the closest distance between ROC curve and the top left corner of the panel.

Specifically, in an iteration I of the forward feature construction process (Supplementary Fig. 3a right section), let $S_{i-1} = [s_1, s_2, \ldots, s_{i-1}]$ be the combination of features selected by the previous $i-1$ iterations, and let $C_{i-1} = [c_i, c_{i+1}, \ldots, c_n]$ be the remaining candidate features. Our algorithm will combine $c_i$ with $S_{i-1}$ to form a new candidate combination and evaluate the performance of this combination by 100 times cross-validations. Similarly, the algorithm will combine $c_{i+1}, c_{i+2}, \ldots,$ or $c_n$ with $S_{i-1}$ to form $n-i$ additional candidate combinations, and evaluate the performance of each candidate combination by 100 times cross-validations. Among these $n-i+1$ candidate combinations, the combination that shows the best performance will be the combination $S_i$ selected by iteration $i$.

**Training CIGdiscover**. Logistic regression model is built on the base of the Sklearn logistic regression library. Data are centralized and normalized using the Pre-processing library in Sklearn. Cross-validation is repeated 100 times by splitting the data into 80% training and 20% test data. Penalty is set as L1 to prevent overfitting in all experiments.

We can denote the response for case $i$ as $y_i$, the $j^{th}$ predictor for case $i$ as $x_{ij}$, the regression coefficient and the intercept corresponding to the $j^{th}$ predictor as $\beta_j$ and $\mu$. Let $\theta = (\mu, \beta_1, \ldots, \beta_p)^t$ and $x_i = (x_i = (x_{i1}, \ldots, x_{ip})^t$, we estimate the parameter vector $\theta$ by maximizing the log-likelihood

$$L(\theta) = \sum_{i=1}^{n} [y_i \log(p_i) + (1 - y_i) \log(1 - p_i)] \quad (1)$$

The Lasso method is implemented by fixing an upper bound on the sum of the absolute value of the model parameters, which can be denoted by penalizing the negative log-likelihood with $L_1$-norm. In the Logistic regression model, the negative log-likelihood is denoted by

$$-\sum_{i=1}^{n} \log\left(p_\beta(y_i|x_i)\right) = \sum_{i=1}^{n} \left\{ -y_i \left(\sum_{j=0}^{p} \beta_j x^{(j)}\right) + \log\left(1 + \exp\left(\sum_{j=0}^{p} \beta_j x^{(j)}\right)\right) \right\} \quad (2)$$

The loss function $\rho$ can be written as

$$\rho_{(\beta)}(x, y) = -y\left(\sum_{j=0}^{p} \beta_j x^{(j)}\right) + \log\left(1 + \exp\left(\sum_{j=0}^{p} \beta_j x^{(j)}\right)\right) \quad (3)$$

The Lasso estimator of a Logistic regression model is defined as

$$\hat{\beta}(\lambda) = \text{argmin}_\beta \left( n^{-1} \sum_{i=1}^{n} \rho_{(\beta)}(x_i, y_i) + \lambda \parallel \beta \parallel_1 \right) \quad (4)$$

For each gene, the signed distance to the hyperplane was used as CIG score. The decision threshold (CIG score cutoff) for CIGs were determined by distance to the top left corner of the ROC curve[41].

$P$-value between a pair of ROC curves was calculated by Hanley's method[42]. First, a critical ratio $z$ will be defined as:

$$z = \frac{A_1 - A_2}{\sqrt{\text{SE}_1^2 + \text{SE}_2^2 - 2r\text{SE}_1\text{SE}_2}} \quad (5)$$

where $A_1$ and $\text{SE}_1$ are the observed area under curve and estimated SE of area under curve for ROC curve 1; where $A_2$ and $\text{SE}_2$ are the associated values for ROC curve 2. $r$ represents the correlation between $A_1$ and $A_2$ via querying the table provided in Hanley's method[42]. Two intermediate correlation coefficients are required to calculate $r$. First, $r_{cig}$ is the Pearson correlation between the CIG scores given to CIGs by the two models; $r_{noncig}$ is the Pearson correlation between the CIG scores given to non-CIG genes by the two models. Furthermore, $r$ is acquired by querying the table[42] using $(r_{cig} + r_{noncig})/2$ and $(A_1 + A_2)/2$. SE of the ROC areas are calculated based on the following equation[43].

$$\text{SE} = \sqrt{\frac{A(1-A) + (na-1)(Q_1 - A*A) + (nn-1)(Q_2 - A*A)}{na*nn}} \quad (6)$$

where $A$ is the area under the curve, na and nn are the number of control genes and CIG genes, respectively, and $Q_1$ and $Q_2$ are estimated by: $Q = A/(2-A)$, $Q = 2A * A/(1+A)$. Then this quantity $z$ is referred to tables of normal distributions and used to estimate $p$-value between the two ROC curves.

**Correlation and collinearity test**. Spearman correlation coefficient between features was calculated using the Python Pandas library. For collinearity analysis, variance inflation factor is calculated using the SciPy library.

**Analyze the influence of cell types**. For cross test, identity genes from only five randomly selected cell types were used to train the model and identity genes from the other cell types were used to test the model. Overlapped genes are removed from the training and test datasets. Similarly, to analyze how the number of cell types influence the model, identity genes that were used for training and for testing were from different cell types. For parallel test, genes that were used for training and testing were from the same cell types.

**Analyze the influence of noises**. To test the robustness of the model, labels of genes were swapped in four different ways: only swap identity genes to negative control genes (false negative), only swap negative control genes to identity genes (false positive), swap equal number of identity genes to control genes and control genes to identity genes (bidirectional false), and randomly change the labels of genes. After swapping, the genes were subject to CIGdiscover to test its performance.

**Training CIGnet**. Data related to network nodes (genes), edges, and closeness between nodes were downloaded from CellNet website (http://cellnet.hms.harvard.edu/)[1]. For each cell-type-specific cell identity subnetwork, only the CIGs were used. Known master transcription factors were defined as in Fig. 1. Control transcription factors are randomly selected from all known transcription factors, except the master transcription factors. Due to the small number of master transcription factors in the training datasets, SMOTE is used to expand the positive genes in training datasets following the distribution of feature values of the known

master transcription factors. All performance tests for the logistic regression model are conducted in the same way as described for CIGdiscover. Cell-type specificity is calculated using Tau index[44] and scaled to be between −1 and 1.

**Pathway analysis**. Top 500 identity genes ranked by the CIG score are selected to perform the DAVID pathway analysis (DAVID 6.8 https://david.ncifcrf.gov/)[45]. The pathways with q-values (adjusted P-values using the Benjamini method) smaller than 0.05 were defined as significantly enriched.

**Cell lineage hierarchy analysis**. For heatmap and hierarchy cluster analysis, we retrieved top 500 identity genes for each cell or tissue type to create a heatmap of cell identity score using the tool Morpheus (https://software.broadinstitute.org/morpheus) with default parameters.

**Data visualization**. Decision boundary analysis is performed using Sklearn and visualized by Matplotlib. ROC curves are created using Matplotlib. Bar plots is created using the Prism statistical software package (Graph Pad Software, Inc., La Jolla, CA, USA). Scatter plots are created using Microsoft Excel software.

Average density of epigenetic marks in promoter region around TSS were plotted using the Profile function in DANPOS version 2.2.3:

python danpos.py profile wig --genefile_paths *putative_identity_genes.txt, putative_negative_identity_genes.txt* --genefile_aliases *positive,negative* --heatmap 1 --name *outdir* --genomic_sites TSS --flank_up 3000 --flank_dn 10000

Heatmap for density of epigenetic marks around TSS is plotted using the software MeV[46] version 4.8.1.

We have added figures for visualization of CIG networks for individual cell or tissue types in the "network visualization section" of our CIGDB at https://sites.google.com/view/cigdb/predicted-db/network-visualization

**Maintenance of human PSCs**. Human PSCs were maintained on Matrigel in mTesR1 medium. Cells were passaged approximately every 6 days. To passage PSCs, cells were washed with Dulbecco's modified Eagle's medium (DMEM)/F12 medium (no serum) and incubated in 1 mg/ml dispase until colony edges started to detach from the dish. The dish was then washed three times with DMEM/ F12 medium. After the final wash, colonies were scraped off of the dish with a cell scraper and gently triturated into small clumps and passaged onto fresh Matrigel-coated plates.

**Human PSCs differentiation to ECs**. Differentiation is induced 4 days after PSCs passaging (day 0). Mesoderm specification is induced by the addition of bone morphogenetic protein 4, activin A, small-molecule inhibitor of glycogen synthase kinase-3β (CHIR99021), and vascular endothelial growth factor (VEGF). Mesoderm inductive factors are removed on day 3 of differentiation and are replaced with vascular specification medium supplemented with VEGF and the transforming growth factor-β pathway small-molecule inhibitor SB431542. Vascular specification medium is additionally refreshed on days 7 and 9 of differentiation. Flow cytometric analysis of differentiated ECs is performed on day 10.

**CRISPR gRNA and lentiviral vector design**. Two open-access software, Cas-Designer (http://www.rgenome.net/cas-designer/) and CRISPR design (http://crispr.mit.edu/), were used to design guide RNAs (gRNA) targeted to candidate gene. Two guides were designed per gene shown as in Supplementary Data 4.

Target DNA oligos were purchased from IDT (Integrated DNA Technologies) and cloned into the lentiCRISPR v2 plasmid[2] (Addgene plasmid# 52961) via BsmBI restriction enzyme sites upstream of the scaffold sequence of the U6-driven gRNA cassette. All plasmids were sequenced to confirm successful ligation.

**Lentiviral constructs**. Lentivirus was packaged by co-transfection of constructs with second-generation packaging plasmids pMD2.G, psPAX2 into a six-well plate with HEK293T cells. After the first 24 h of transfection (250 ng of pMD2.G, 750 ng of psPAX2, 1 μg of target plasmid), the medium was changed to DMEM and the supernatants 48 and 72 h after transfection were pooled, filtered through a 0.45 μm filter, and used for infection.

**Cell culture and lentiviral transduction**. HUVECs were purchased from Lonza (C2517A) and human pluripotent stem cell (hPSC) was a kind gift from Dr John Cooke's group. All cells used in this study were within 15 passages after receipt. HUVECs were cultured in 5% $CO_2$ and maintained in vitro in Endothelial Growth Basal Medium with EGM-2 SingleQuot Kit. hPSC was cultured in 5% $CO_2$ and maintained in vitro in mTeSR1 basal medium with mTeSR1 supplement. Those cell lines were mycoplasma negative during routine tests. HUVECs were grown to 70% confluence and infected with lentiviral vectors. The media was changed 8 h after viral transduction and incubated for 48 h before selection with 1 μg/mL puromycin for 3 days. HUVECs were collected for cell proliferation assay, nitric oxide production, and genomic DNA extraction. On the second day of PSC passaging, PSCs were infected with lentiviral vectors. The media was changed 6 h after viral transduction. Differentiation was induced 3 days after virus infection.

**T7 endonuclease I assay**. Genomic DNA from lentiviral transduced cells were extracted with a Quick-DNA Miniprep Kit (Zymo Research, Irvine, CA) following manufacturer's protocol and were quantified using a Synergy 2 Multi-Mode Reader (BioTek, Winooski, VT, USA). The targeted regions were PCR-amplified with amfiSure PCR Master Mix (GenDEPOT, Barker, TX, USA) using primers flanking the target sites. Primers sequence are shown in Supplementary Data 4.

We denatured 200 ng of the PCR products and then slowly hybridized to form heteroduplexes using the following program settings: 95 °C for 5 min, 95°–85 °C at −2 °C/s, 85°–25 °C at −0.1 °C/s. Heteroduplexes were digested with T7 endonuclease I (New England Biolabs, Ipswich, MA, USA) at 37 °C for 30 min. In addition, the digested products were separated on a 2% TAE agarose gel for analysis. Images were captured using the ChemiDoc XRS+ Molecular Imager system (Bio-Rad, Hercules, CA, USA).

**Cell proliferation assay**. After viral transduction and puromycin selection, HUVECs were plated in 96-well plates at a density of 1000 cells per well and allowed to attach for 24 h. Viability was measured utilizing the CellTiter-Glo® Luminescent Cell Viability Assay (Promega, Madison, WI, USA). Results were read at 24, 48, and 72 h on the on Synergy 2 Multi-Mode Reader.

**Nitric oxide production assay**. HUVECs ($8 \times 10^3$) were added to triplicate wells of a 96-well plate with 200 ml media. 4-amino-5methylamino-2979-difluorofluorescein diacetate (10 mM) in anhydrous dimethylsulfoxide was added to each well and the plate was incubated for 30 min at 37 °C and 5% $CO_2$. The cells were washed with PBS, 200 ml fresh media was added, and the plate was incubated for an additional 30 min. Fluorescence was measured using a Synergy 2 Multi-Mode Reader.

**Endothelial cell tube formation assay**. Ninety-six-well plates were coated with 50 μl Matrigel (R&D Systems, catalog number 3432-005-01) and incubated at 37 °C for 30 min. Control ($1 \times 10^4$) and CRISPR/Cas9-edited HUVECs in 100 μl EGM medium were seeded in each well, respectively. After 4 h, images were captured using the Leica epi-fluorescence microscope. Branches number and length were quantified using ImageJ software.

**Fluorometric LDL uptake assay**. Control and CRISPR/Cas9-edited HUVECs were seeded on a 48-well plate. Alexa Fluor 594 AcLDL (Thermo Fisher Scientific, catalog number L35353) was added to the culture medium for the final 4 h of the incubation time. HUVECs were washed, trypsinized, and centrifuged at $200 \times g$ for 5 min, then resuspended in FACSB-10. Fluorescence was determined using a flow cytometer (LSR II, Becton-Dickinson, San Jose, CA, USA) and the data were analyzed using FlowJo software.

**Flow cytometric analysis**. Ten days after differentiation, PSCs were trypsinized, centrifuged at $200 \times g$ for 5 min, resuspended in FACSB-10 (FACS buffer–10% fetal bovine serum) and incubated with anti-human-VE-cadherin (Invitrogen, catalog number 17-0319-41, 1:150) and anti-human CD31 (Invitrogen, catalog number 53-1449-41, 1:150) for 30 min on ice. Fluorescence was determined using a flow cytometer (LSR II, Becton-Dickinson, San Jose, CA, USA) and the data were analyzed using FlowJo software.

**Statistical analysis**. For bar plots and box plots, P-values are calculated with Wilcoxon's test (two-tail). For ROC curves, P-values are calculated by Hanley's method[42] (two-tail). Q-values (Benjamini) for pathways were directly determined using DAVID (https://david.ncifcrf.gov/). For CRISPR9 experiments, data were presented as mean ± SD of six individual experiments. Statistical analysis was performed with Student's T-test by means of the Prism statistical software package (Graph Pad Software, Inc., La Jolla, CA, USA).

**Reporting summary**. Further information on research design is available in the Nature Research Reporting Summary linked to this article.

## Data availability
Data accession numbers are listed in Supplementary Data 3. Data generated during the study available in a public repository at https://sites.google.com/site/cellidentitygene/. The RNA-seq, H3K4me3, H3K4me1, H3K27ac, and H3K27me3 ChIP-seq data for the ten well-defined cell types and landscape analysis are downloaded from GEO database and ENCODE project website (https://www.encodeproject.org/)[38].

## Code accessibility
The bioinformatics toolkit CEFCIG is available at https://github.com/bxia888/CEFCIG. The catalog of CIGs was deposited to the database CIGDB developed for this project and available at https://sites.google.com/view/cigdb.

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

## Acknowledgements

This work was supported by grants from NIH/NIGMS (R01GM125632 to K.C.) and NIH/NHLBI (R01HL133254 and R01HL148338 to K.C. and J.C.). Q.C. is supported by US Department of Defense (W81XWH-15-1-0639 and W81XWH-17-1-0357), American Cancer Society (TBE-128382), and NIH/NCI (R01CA208257 and Prostate SPORE P50CA180995 DRP). We thank Scientific Writer Dr Johnique T. Atkins for revising the manuscript.

## Author contributions

K.C. conceived the project, designed the experiments, and interpreted the data. B.X., D.Z., G.W., and A.T. performed the data analysis. Q.C. conceived the experiments and interpreted the data. L.Z. and M.Z. designed and performed the experiments and analyzed the data. K.C. Q.C., B.X., L.Z., D.Z., and J.L. wrote the manuscript with comments from A.T., J.L., Y.B., Y.L., S.M., X.W., and J.C.

## Competing interests

The authors declare no competing interests.
