## [Peer Review File · Nature Communications]

Reviewers' comments:

Reviewer #1 (Remarks to the Author):

The manuscript "Machine Learning uncovers cell identify regulator by histone code" by B. Xia et al. is timely in this era of single cell RNA-seq and the need to identify cell type specific genes including master regulators of cell type specific states. The authors provide a suite of software tools to identify cell identify genes (CIGs) from RNA-seq and histone modification ChIP-seq data of marks whose patterns have been observed in previous studies to display sticking difference at CIGs compared to other genes. In particular, H3K4me3, H3K4me1, and H3K27ac have been shown to spread across the body of CIGs as opposed to being localized at their promoters as they are at other expressed genes. The authors curate CIGs and have developed a database CIGDB (which has been submitted as a separate article) which they use to train and validate their CIG predictive models. Among the analyses that makes this a useful set of approaches and tools for investigators is the demonstration that one mark (e.g., H3Kme1 or H3K4me3) is sufficient to yield reasonable predictive power with a boost coming from their GridGO optimization and logistic regression across various inputs which characterize histone modification shape including peak height, width, integrated signal, coverage, skewness and kurtosis compared to just focusing on the width of a given mark's peak across a gene. This is important because most investigators will not necessarily perform ChIP-seq on three histone modifications to address the question of CIGs in their studies. The following suggestions are intended to strengthen the conclusions and make the computational approaches more clear so that others can reproduce the authors' results.

Major comments:

(1) As shown in Fig. 1I, for example, there is an enrichment of known endothelial cell (EC) CIGs including NR2F2 and FOXP1 at the top; however, there are many predicted CIGs (in grey) that are ranked higher than other known EC CIGs including FLI1 and TAL1. Are all of these genes in grey that score higher than FLI1 and TAL1 CIGs? Will other investigators risk performing time consuming, costly experiments based on these predictions? It's not clear given the validation experiments performed. Predicted EC CIGs were validated by using CRISPR-Cas9 KO followed by assessing the impact of these KOs on EC proliferation and NO production. However, the impact on NO production tends to be ~10% (when significant) while the impact on proliferation appears more dramatic. But proliferation is not a unique feature of ECs! How specific is assaying proliferation to the assessment that a given gene is a CIG? Positive and negative controls would be important to address the specificity of the predicted CIGs and are lacking in both the set of experiments whose results are shown in Fig. 1J-L and Fig. 3G-H.

(a) In the case of Fig 1J-L, a possible set of negative controls include CRISPR-Cas9 of some of the same CIGs/TFs (e.g., GATA3, MECOM, etc) in other cell types that proliferate and in which the same CIGs/TFs are expressed and demonstrate no significant effect on proliferation of knocking out these same factors in another cell type. A set of positive controls would be to knock out a known EC CIGs/TFs and assess their impact on proliferation and NO production. Is the impact of the known CIGs/TFs stronger, the same or even weaker than the predicted set of CIGs/TFs? One would hope they're comparable.

(b) In the case of Fig 3G-H, assaying the efficiency of endothelial cell induction in WT and CIG/TF KO cells more directly addresses the role of CIGs in cell type identity. It would be helpful to KO (using the same CRISPR-Cas9 approach) known endothelial cell CIGs/master TFs (positive controls) as well as expressed TFs that are not CIGs and have a low predicted CIG score (negative controls). Again, Is the impact of the known CIGs/TFs stronger, the same or even weaker than the predicted set of CIGs/TFs? One would hope they're comparable. Is a 20-60% reduction in induction efficiency consistent with KO of a known master TF/CIG?

(2) The authors should calculate and report balanced measures of specificity and sensitivity including positive predictive value (or precision), recall and negative predictive value for just their

top ROC curves: CIGdiscover/red line in Fig 1F and CIGNet/red line in Fig 3E. They select the same number of random negative control CIGs so this was balanced; nevertheless, these other measures will help calibrate the ROC curves and give a further indication of the specificity and sensitivity of the predictive models.

(3) The description of the GRIDGO algorithm in the Methods section (only place it appears) and Fig S2A don't match each other. The authors should expand both their description of the GRIDGO method to include the flow chart shown in Fig S2A and expand Fig S2A to include the approach of iteratively "zooming in" on grids that have the most significant p-values until the minimum grid size is reached. Both the figure and description in the Methods section should be sufficiently clear that someone could attempt to reproduce the approach.

(4) The formula's used for logistic regression including the L1 penalty should be shown in the section Training CIGdiscover in the Methods section. The authors should also describe which detailed approaches that they took using Hanley's method to estimate p-values for their ROC curves. Hanley's paper describes a number of approaches to estimating the standard error of the ROC AUC, for example. Enough detail should be provided that a reader could calculate and reproduce the reported p-values.

(5) Regarding the Training CIGnet section in Methods, the authors should generate a few illustrative images (which they can put in their supplement) of the networks that they generated and define their network parameters for biologists. The summaries shown are no question the correct ones for making conclusions, but as presented, the networks are extremely abstract and just summarized statistically. Notably, networks cannot even be viewed on-line by going to the <http://cellnet.hms.harvard.edu> site as the web application is down! Most readers will not go through the trouble of downloading and running the code in order to visualize the networks used in this manuscript.

(6) The authors should specify a clear criteria for selecting CIGs using their predictive models (maybe I missed this but could not see it). They currently demonstrate high enrichment for known CIGs among predicted ones with high CIG scores, but no explicit criteria or cutoff method for investigators who would be interested in using their tools/approaches.

Minor comments:

(1) There are a few sentences where the claims are a bit too strong and should be "softened" including line 289 "We delivered a foundation of knowledge, and two paradigm shift techniques..."; line 262 "Our methods developed in this work are conceptually novel..." they are more precise but previous groups defined the concepts of using broad/unusual histone modification patterns to identify CIGs; and in lines 312-313 "...CIGnet predicts master transcription factors with exceptional performance." As the authors indicate and this reviewer checked, there is a lack of explicit computational methods developed for the goal of identifying CIGs from histone modification data.

(2) In the CHIP-seq and RNA-seq data analysis section of the Methods section, the authors show code that they run to generate data that would be used for further downstream analysis, but some important details were lacking. For example, was quantile normalization performed on the RNA-seq and CHIP-seq data? Were all RNA-seq data sets quantile normalized across cell types? Was this also the case for each histone modification CHIP-seq data set across samples?

(3) While relatively well written, it could be improved by careful proofreading for grammar and style.

Best,
Stefan Bekiranov

Reviewer #2 (Remarks to the Author):

This work by Xia et al. proposed a machine learning approach for identifying cell identity genes (CIGs) using histone marks. The authors selected 10 top-ranked cell types from 297 candidates based on the associated PubMed abstracts and availability of RNA-seq and ChIP-seq of H3K4me1/3 and H3K27ac/me3. The authors next created a set of 6 features for each histone modification marks. Together with RNA-seq, this led to 49 features that the authors performed feature selection on for identifying CIGs. To this reviewer, it is not clear how the proposed approach differs from previous studies. Furthermore, there appear to be several pitfalls in the experimental design and evaluation procedure.

1. A number of previous studies (e.g. "Cismapper: predicting regulatory interactions from transcription factor ChIP-seq data" (PMC5389714); and "A predictive modeling approach for cell line-specific long-range regulatory interactions" (PMC4605315)) have proposed to machine learning from histone modification marks (among other genomic features) for predicting enhancers and/or transcription factor target genes that are cell type-specific. The cell identity genes referred to in this study appears to be conceptually the same set of genes as cell type-specific genes (i.e. the four categories of genes that the author defined for CIGs are also cell type-specific genes (page 1; line 90-94)). Since the authors didn't discuss the difference of their work in the context of these previous work, the utilization of machine learning from histone marks for CIG prediction does not seem to bring any conceptual novelty to the field.

2. Figure 1F shows that by using a single feature such as RNA expression (or H4K4me3 width etc.), the performance of the prediction is already very good. However, for example, when only using RNA expression data, the genes are simply ranked by their expression levels. How could such an approach be able to achieve the performance as shown in Figure 1F? Are CIGs a subset of highly expressed genes? Equally interesting is that a single histone feature can also lead to similar prediction accuracy as RNA expression. I found this hard to believe. What is the underpinning biology supporting this?

3. Given the above observation, the definition of negative control genes does not appear to properly control for CIGs. Specifically, for a given cell type, the authors randomly selected from genes that do not appear with that cell type in literature and other major databases (page 11; line 481-486). Control genes selected this way will most likely to display much lower expression levels control to CIGs because CIGs are by definition expressed in the cell type of consideration. Given the high prediction power shown in Figure 1F from using RNA expression alone, the authors should at least select control genes that have similar levels of expression to CIGs.

4. The number of CIGs curated for each cell type is relatively small (Figure 1A and B). Clearly, there are a large number of CIGs that are not curated for each cell type and they are what the authors attempt to predict for. Then, how would the authors be sure that when they randomly select for control genes for CIGs, these randomly selected genes themselves are not CIGs? The author should take this into account when selecting for control genes, so the learning model will not over penalize potential CIGs because some of the unknown CIGs were accidentally selected as control genes.

5. A grid-based genetic method (GridGO) was used for finding the optimal ChIP-seq peak height, boundary, and several other characteristics that can improve the prediction accuracy of CIG. This procedure does not appear to be nested in any cross-validation procedure. As such, the optimization of ChIP-seq parameters will lead to overfitting because the histone features are optimized for CIG prediction using training data prior to the feature selection and prediction steps.

6. The feature selection step appears to be nested in a random split of training and test procedure (as depicted in Figure S3A). However, it is not clear why the 9 features (Figure 1D) were always

selected from the forward feature selection procedure. Due to the random split of the training and test data, different number and different combination of features are expected to be found.

7. The authors found that the combination of H3K4me1/3 and H3K27ac can accurately identify CIGs within and across multiple cell types (Figure S5A) and across the four categories of CIGs (Figure S5D). This implies that the same combination of H3K4me1/3 and H3K27ac features define CIGs in all cell types considered in this study. It is known that the presence of H3K4me1/3 and H3K27ac marks are associated with active genes. How could such a combination be specific to CIGs and at the same time be non-cell-type-specific is surprising and at the same time hard to understand. Could this be an artifact caused by overfitting per my comments in (5) and (6)?

8. There are multiple ways to calculate a specificity score (PMC5444245) from multiple cell types given the RNA expression data. Would the specificity score calculated from using alternative methods (alternative to the "Tau" method used in (Figure S7B)) give equal or better performance in identifying CIGs than such a machine learning approach? The authors should perform a "cross-validation" by using different specificity scores to quantify the performance and compare their machine learning predicted genes with genes found by an alternative specificity metric to the one that is used for quantifying the performance.

We thank all reviewers for reading our manuscript very carefully, for acknowledging the novelty and significance of our work, and for providing many constructive comments. In the revised manuscript, we have performed a number of new experiments and analyses suggested by each reviewer, as indicated by Figures 1I, 1H, 3G, S2A, S3A, S5-S10, S11C, and the texts highlighted in yellow color in the new manuscript. We are confident that the revised manuscript has been substantially improved to fit the high quality of manuscript for *Nature Communications*.

Reviewers' comments:

Reviewer #1 (*Remarks to the Author*):

The manuscript "Machine Learning uncovers cell identify regulator by histone code" by B. Xia et. al. is timely in this era of single cell RNA-seq and the need to identify cell type specific genes including master regulators of cell type specific states. The authors provide a suite of software tools to identify cell identify genes (CIGs) from RNA-seq and histone modification ChIP-seq data of marks whose patterns have been observed in previous studies to display sticking difference at CIGs compared to other genes. In particular, H3K4me3, H3K4me1, and H3K27ac have been shown to spread across the body of CIGs as opposed to being localized at their promoters as they are at other expressed genes. The authors curate CIGs and have developed a database CIGDB (which has been submitted as a separate article) which they use to train and validate their CIG predictive models. Among the analyses that makes this a useful set of approaches and tools for investigators is the demonstration that one mark (e.g., H3Kme1 or H3K4me3) is sufficient to yield reasonable predictive power with a boost coming from their GridGO optimization and logistic regression across various inputs which characterize histone modification shape including peak height, width, integrated signal, coverage, skewness and kurtosis compared to just focusing on the width of a given mark's peak across a gene. This is important because most investigators will not necessarily perform ChIP-seq on three histone modifications to address the question of CIGs in their studies. The following suggestions are intended to strengthen the conclusions and make the computational approaches more clear so that others can reproduce the authors' results.

A: We thank Dr. Bekiranov for highlighting that our work “is timely in this era of single cell RNA-seq and the need to identify cell type specific genes including master regulators of cell type specific states”, “among the analyses that makes this a useful set of approaches and tools for investigators is the demonstration that one mark (e.g., H3Kme1 or H3K4me3) is sufficient to yield reasonable predictive power with a boost coming from their GridGO optimization and logistic regression”, and “this is important because most investigators will not necessarily perform ChIP-seq on three histone modifications to address the question of CIGs in their studies”. We would like to particular appreciate our reviewer for providing suggestions “intended to strengthen the conclusions and make the computational approaches more clear”.

Major comments:

(1) As shown in Fig. 1I, for example, there is an enrichment of known endothelial cell (EC) CIGs including NR2F2 and FOXP1 at the top; however, there are many predicted CIGs (in grey) that are ranked higher than other known EC CIGs including FLI1 and TAL1. Are all of these genes in grey that score higher than FLI1 and TAL1 CIGs? Will other investigators risk performing time consuming, costly experiments based on these predictions?

A: Although it is impossible in this manuscript for us to experimentally verify all the predicted cell identity genes (CIGs), our current results suggest high probability that all

of these genes are CIGs. (1) Figure 1F indicate that our algorithm recapture known CIGs with high accuracy (sensitivity 0.89, specificity 0.92, precision 0.91, negative predictive value 0.90). (2) Comprehensive experimental verifications indicate that our algorithm define new CIGs with high accuracy. All 8 tested candidates show significant effects on at least two of four tested EC phenotypes, including proliferation, NO production, LDL uptake and tube formation. (3) Although we only highlighted 5 EC CIGs in the figure due to space limitation, our thorough literature review revealed that 255 (42.9%) of the predicted CIGs for EC have reported role in endothelial differentiation, phenotypes, or functions (**Table S2**). Therefore, when researchers in the community choose a gene from our predicted list to investigate its endothelial role, the success rate would be also high and thus the risk will be low.

It's not clear given the validation experiments performed. Predicted EC CIGs were validated by using CRISPR-Cas9 KO followed by assessing the impact of these KOs on EC proliferation and NO production. However, the impact on NO production tends to be ~10% (when significant) while the impact on proliferation appears more dramatic. But proliferation is not a unique feature of ECs! How specific is assaying proliferation to the assessment that a given gene is a CIG? Positive and negative controls would be important to address the specificity of the predicted CIGs and are lacking in both the set of experiments whose results are shown in Fig. 1J-L and Fig. 3G-H.

A: We agree that proliferation is not a unique feature of ECs, although this feature is particularly important to EC because EC proliferation is key to angiogenesis. Therefore, in addition to the EC proliferation assay, we performed NO production assay. More importantly, we now have further performed LDL uptake and tube formation assays (see responses to a and b) in the revised manuscript. Following our reviewer's suggestions, we have added positive and negative control genes for all verification experiments (see responses to a and b).

(a) In the case of Fig 1J-L, a possible set of negative controls include CRISPR-Cas9 of some of the same CIGs/TFs (e.g., GATA3, MECOM, etc) in other cell types that proliferate and in which the same CIGs/TFs are expressed and demonstrate no significant effect on proliferation of knocking out these same factors in another cell type. A set of positive controls would be to knock out a known EC CIGs/TFs and assess their impact on proliferation and NO production. Is the impact of the known CIGs/TFs stronger, the same or even weaker than the predicted set of CIGs/TFs? One would hope they're comparable.

A: (1) Following the suggestion, we tested the effect of four EC identity genes on fibroblast proliferation. Although three of these four genes expressed in fibroblast (**Fig R1A**), their knockouts in fibroblast did not show significant effect on fibroblast proliferation (**Fig R1B**). (2) For additional negative controls, we selected six genes that expressed in EC (**Fig S5**) but were not predicted to be EC CIGs, and tested their effects on EC phenotypes. We found none of these six negative control genes show significant effect on EC proliferation (**Figure S7**), NO production (**Figure S8**), LDL uptake (**Figure S9**), or EC tube formation (**Figure S10**). (3) We also tested two positive control genes that expressed in EC (**Figure S5**). Although all 8 tests (4 phenotypes x 2 genes) for positive control genes appeared significant, whereas 27 (84.38%) of the 32 tests (4 phenotypes x 8 genes) for predicted CIGs were significant (**Figure 1J**), we found the predicted EC CIGs and the positive control genes show overall comparable effects on individual EC phenotypes (**Figure S7-10**).

(b) In the case of Fig 3G-H, assaying the efficiency of endothelial cell induction in WT and CIG/TF KO cells more directly addresses the role of CIGs in cell type identity. It would be helpful to KO (using the same CRISPR-Cas9 approach) known endothelial cell CIGs/master TFs (positive controls) as well as expressed TFs that are not CIGs and have a low predicted CIG score (negative controls). Again, Is the impact of the known CIGs/TFs stronger, the same or even weaker than the predicted set of CIGs/TFs? One would hope they're comparable. Is a 20-60% reduction in induction efficiency consistent with KO of a known master TF/CIG?

A: In the revised manuscript, we have tested two positive control genes and two negative control genes that expressed in EC (**Fig S5**). We observed that knockouts of the predicted CIGs and knockouts of the positive control genes did cause comparable reduction in EC induction efficiency, whereas knockouts of the two expressed negative control genes have no significant effect on the induction efficiency (**Fig 3G**).

(2) The authors should calculate and report balanced measures of specificity and sensitivity including positive predictive value (or precision), recall and negative predictive value for just their top ROC curves: CIGdiscover/red line in Fig 1F and CIGNet/red line in Fig 3E. They select the same number of random negative control CIGs so this was balanced; nevertheless, these other measures will help calibrate the ROC curves and give a further indication of the specificity and sensitivity of the predictive models.

A: We appreciate Dr. Bekiranov for highlighting that we selected the same number of random negative control genes so that our models are balanced. We have further measured the specificity, sensitivity, positive predictive value (precision), and negative predictive value for CIGdiscover and CIGNet. These values have been added to the revised manuscript as “The CIGdiscover successfully recaptured known identity genes, with a sensitivity value 0.89, specificity value 0.92, precision value 0.91, negative predictive value 0.90” and “CIGnet predict master transcription factors of CIGs with a sensitivity value 0.88, specificity value 0.88, precision value 0.88, negative predictive value 0.89”.

(3) The description of the GRIDGO algorithm in the Methods section (only place it appears) and Fig S2A don't match each other. The authors should expand both their description of the GRIDGO method to include the flow chart shown in Fig S2A and expand Fig S2A to include the approach of iteratively "zooming in" on grids that have the most significant p-values until the minimum grid size is reached. Both the figure and description in the Methods section should be sufficiently clear that someone could attempt to reproduce the approach.

A: We appreciate Dr. Bekiranov for this suggestion. We have revised the **Figure S2A** accordingly. The description of GridGO in the methods section has been revised as: “We developed GridGO, a grid-based genetic method to optimize bioinformatics parameters for detecting epigenetic signature of CIGs. We use GridGO to optimize 3 important parameters, including the height cutoff to define ChIP-Seq enrichment peak, the upstream distance cutoff to assign a peak to a nearby gene, and the downstream distance cutoff to assign a peak to a nearby gene. However, GridGO is designed to allow optimizing different number of parameters. For simplicity, we will describe details of the algorithm by an example in which the upstream and downstream distance cutoffs to assign a ChIP-seq peak to nearby genes are set to be the same, so that GridGO will optimize only two parameters including the height cutoff to define ChIP-Seq enrichment peak and the distance cutoff to assign a peak to a nearby gene (**Fig S2A**) . In the first iteration of optimization, the entire two-dimensional parameter space was divided into m equal-size grids. Then the parameter values in the center of each grid were used to

define ChIP-Seq enrichment peaks and to assign the peaks to nearby genes. Afterwards, P value of difference in a peak feature (epigenetic signature) between CIGs and control genes were determined by Wilcoxon test. The grid with the lowest P value was the optimal grid saved for the second iteration. In the second iteration, the grid saved in the first iteration was divided into a new set of m small grids, which will be tested as previous iteration to select an optimal grid saved for the third iteration. Such genetic evolution of parameter grid kept going until the number of iteration become larger than a given value n or the new optimal grid was not better than the previous optimal grid. To estimate potential over-fitting effect, only 80% of training genes were used in the GridGo optimization and the CIGdiscover model was built based on parameters optimized by these genes. Then the performance of CIGdiscover on these 80% genes and the rest 20% genes were compared, and little over-fitting effect was observed.”

(4) The formula's used for logistic regression including the L1 penalty should be shown in the section Training CIGdiscover in the Methods section. The authors should also describe which detailed approaches that they took using Hanley's method to estimate p-values for their ROC curves. Hanley's paper describes a number of approaches to estimating the standard error of the ROC AUC, for example. Enough detail should be provided that a reader could calculate and reproduce the reported p-values.

A: The formulas used for logistic regression including the L1 penalty and Hanley's method to estimate p-values for ROC curves have been added into the revised manuscript as:

“We can denote the response for case i as y_i , the j^{th} predictor for case i as x_{ij} , the regression coefficient and the intercept corresponding to the j^{th} predictor as β_j and μ . Let $\theta = (\mu, \beta_1, \dots, \beta_p)^t$ and $x_i = (x_{i1}, \dots, x_{ip})^t$, we estimate the parameter vector θ by maximizing the loglikelihood

$$L(\theta) = \sum_{i=1}^n [y_i \log(p_i) + (1 - y_i) \log(1 - p_i)]$$

Lasso penalized regression is implemented by minimizing the cost function

$$g(\theta) = \sum_{i=1}^n (y_i - \mu - x_i^t \beta)^2 + \lambda \sum_{j=1}^p |\beta_j|.$$

For each gene, the signed distance to the hyperplane was used as CIG score. The decision threshold (CIG score cutoff) for CIGs were determined by distance to top left-corner of ROC curve¹.

P value between a pair of ROC curves was calculated by Hanley's method². First, a critical ration z will be defined as:

$$z = \frac{A_1 - A_2}{\sqrt{SE_1^2 + SE_2^2 - 2rSE_1SE_2}},$$

where A_1 and SE_1 are the observed area under curve and estimated standard error of area under curve for ROC curve 1; where A_2 and SE_2 are the associated values for ROC curve 2. r represents the correlation between A_1 and A_2 via querying the table provided in Hanley's method². Two intermediate correlation coefficients are required to calculate r . First, r_{cig} , is the Pearson correlation between the CIG scores given to CIGs by the two models; r_{noncig} , is the Pearson correlation between the CIG scores given to non-CIG genes by the two models. And then r is acquired by querying the table² using $(r_{\text{cig}} +$

$r_{noncig}/2$ and $(A_1 + A_2)/2$. Standard error of the ROC areas are calculated based on the following equation ³:

$$SE = \sqrt{\frac{A(1 - A) + (na - 1)(Q_1 - A * A) + (nn - 1)(Q_2 - A * A)}{na * nn}},$$

where A is the area under the curve, na and nn are the number of control genes and CIG genes respectively, and Q_1 and Q_2 are estimated by: $Q = A/(2-A)$, $Q = 2A*A/(1+A)$. Then this quantity z is referred to tables of normal distributions and used to estimate p value between the two ROC curves.”

(5) Regarding the Training CIGnet section in Methods, the authors should generate a few illustrative images (which they can put in their supplement) of the networks that they generated and define their network parameters for biologists. The summaries shown are no question the correct ones for making conclusions, but as presented, the networks are extremely abstract and just summarized statistically. Notably, networks cannot even be viewed on-line by going to the <http://cellnet.hms.harvard.edu> site as the web application is down! Most readers will not go through the trouble of downloading and running the code in order to visualize the networks used in this manuscript.

A: We appreciate Dr. Bekiranov for highlighting that “*The summaries shown are no question the correct ones for making conclusions*”. Following our reviewer’s suggestion, we have added figures for the associated CIG networks in the “network visualization section” of our Cell Identity Gene Data Base (CIGDB) at: <https://sites.google.com/view/cigdb/predicted-db/network-visualization>

(6) The authors should specify a clear criteria for selecting CIGs using their predictive models (maybe I missed this but could not see it). They currently demonstrate high enrichment for known CIGs among predicted ones with high CIG scores, but no explicit criteria or cutoff method for investigators who would be interested in using their tools/approaches.

A: We used the distance to top left-corner of ROC curve to determine the decision threshold for CIG scores¹. We added this criterion into the revised Method section as “The decision threshold (CIGs score cutoff) for CIGs were determined by distance to top left-corner of ROC curve”

Minor comments:

(1) There are a few sentences where the claims are a bit too strong and should be "softened" including line 289 "We delivered a foundation of knowledge, and two paradigm shift techniques..."; line 262 "Our methods developed in this work are conceptually novel..." they are more precise but previous groups defined the concepts of using broad/unusual histone modification patterns to identify CIGs; and in lines 312-313 "...CIGnet predicts master transcription factors with exceptional performance." As the authors indicate and this reviewer checked, there is a lack of explicit computational methods developed for the goal of identifying CIGs from histone modification data.

A: Thank you for these suggestions. These parts were softened in the new manuscript to be more precise, including “We developed a manually curated database of reported cell identity genes, and further developed two machine-learning models for prediction of new cell identity genes”, “we developed the machine-learning model CIGdiscover that determines the optimal combination of epigenetic signatures to define cell identity genes”, and “CIGnet predicts master transcription factors on the basis of calculated importance

of individual network features”. Some other writings in the manuscript were also softened to be more precise.

(2) In the ChIP-seq and RNA-seq data analysis section of the Methods section, the authors show code that they run to generate data that would be used for further downstream analysis, but some important details were lacking. For example, was quantile normalization performed on the RNA-seq and ChIP-seq data? Were all RNA-seq data sets quantile normalized across cell types? Was this also the case for each histone modification ChIP-seq data set across samples?

A: All RNAseq datasets were performed with quantile normalization across cell types. For ChIP-seq datasets, quantile normalization was performed using DANPOS2.2.3 across cell types. We added these details to the revised manuscript by “Afterwards, quantile normalization of gene expression values was performed across cell and tissue types” and “By this method, ChIP-Seq data from different cell and tissue types were all normalized to have the same quantiles”. Some other descriptions in the Methods section are also revised to be more accurate.

(3) While relatively well written, it could be improved by careful proofreading for grammar and style.

A: We invited Dr. Johnique T. Atkins, a professional Scientific Writer for our department, to improve the manuscript on grammar and style.

Reviewer #2 (Remarks to the Author):

This work by Xia et al. proposed a machine learning approach for identifying cell identity genes (CIGs) using histone marks. The authors selected 10 top-ranked cell types from 297 candidates based on the associated PubMed abstracts and availability of RNA-seq and ChIP-seq of H3K4me1/3 and H3K27ac/me3. The authors next created a set of 6 features for each histone modification marks. Together with RNA-seq, this led to 49 features that the authors performed feature selection on for identifying CIGs. To this reviewer, it is not clear how the proposed approach differs from previous studies. Furthermore, there appear to be several pitfalls in the experimental design and evaluation procedure.

A: We thank our reviewer for many thoughtful comments and questions. We summarized our reply into two major points:

1) Cell identity genes are not conceptually the same as cell type-specific genes. Although some cell identity genes can be very specific to one cell type, some other cell identity genes might be expressed in related cell types and have different degrees of expression specificity. Further, expression specificity analysis requires comparison between a target cell type and most, if not all, other cell types. It also requires distinguishing between cell-type-specific and biological-condition-specific genes, e.g., heat shock genes. Therefore, it is not cost-effective, if not impossible, to collect data for all cell types under all biological conditions for the expression specificity analysis. As such, it is inaccurate to define cell identity genes by expression specificity analysis, as described in our response to the major point #1 (**Figure R4**).

2) Our new method for identification of cell identity genes is distinct from previous methods that were based on expression specificity analysis, because our method relies primarily on the difference in epigenetic mechanism for transcriptional regulation between cell identity genes and other active genes in the same cell type. It does not require comparison of gene expression between cell types. Epigenetic markers not only indicate activation or repression of a gene, but also, further indicate the different ways in which different genes are activated or repressed. For example, although both housekeeping genes and cell identity genes are activated in a cell, housekeeping genes have sharp enrichment of H3K4me3 in promoter, whereas cell identity genes have broad H3K4me3 on both promoter and gene body⁴. This is the reason why we could distinguish cell identity genes from other active genes in the same cell type without a comparison to other cell types.

1. A number of previous studies (e.g. "Cismapper: predicting regulatory interactions from transcription factor ChIP-seq data" (PMC5389714); and "A predictive modeling approach for cell line-specific long-range regulatory interactions" (PMC4605315)) have proposed to machine learning from histone modification marks (among other genomic features) for predicting enhancers and/or transcription factor target genes that are cell type-specific. The cell identity genes referred to in this study appears to be conceptually the same set of genes as cell type-specific genes (i.e. the four categories of genes that the author defined for CIGs are also cell type-specific genes (page 1; line 90-94)). Since the authors didn't discuss the difference of their work in the context of these previous work, the utilization of machine learning from histone marks for CIG prediction does not seem to bring any conceptual novelty to the field.

A: We thank reviewer for reminding us to discuss the difference between our work and previous works that define cell type-specific target genes for transcription factor. In the

first work, Dr. Timothy O'Connor and colleagues utilized the correlation between a histone mark at binding sites of a transcription factor and the expression of individual genes to facilitate the prediction of the long-range regulatory relationship between the TF and its target genes. In the second work, Dr. Sushmita Roy and colleagues described an interesting machine learning model to predict long-range chromatin interactions based on histone modifications. By pairwise comparison between predicted 3D interactions from different cell types, the study discovered many cell type specific interactions. Both studies provided important information to advance our understanding of tissue specific relationships between TF and its target genes, and have been discussed in the revised manuscripts as:

"Recently, epigenetic signatures were used as the inputs of various computational models to define cell type specific regulatory relationship between transcription factors and their target genes on the basis of three dimensional chromatin interactions^{5,6}, expression profiles⁷, and binding of transcription factors⁸. When combined with cell type specific genes, these methods could be further utilized to identify important transcription factors for the associated cell type. Although some cell identity genes can be very specific to one cell type, some other cell identity genes might be expressed in multiple related cell types and thus have different degrees of expression specificity. Therefore, cell identity genes are not conceptually the same as cell type-specific genes. Here, we developed the machine-learning model CIGdiscover that determines the optimal combination of epigenetic signatures to define cell identity genes, and further developed the network model CIGnet to identify the master transcription factors that regulate cell identity genes."

Compare to genes nominated by models that are based on specificity score and use different methods to measure specificity, our CIGdiscover model is the most accurate when used to predict cell identity genes (**Figure R4A**). Further, Genes predicted by CIGdiscover are significantly stronger enrichment in pathways related to cell differentiation (e.g., endothelial cell differentiation) or cell identity (e.g., VEGF signaling) (**Figure R4B**).

2. Figure 1F shows that by using a single feature such as RNA expression (or H4K4me3 width etc.), the performance of the prediction is already very good. However, for example, when only using RNA expression data, the genes are simply ranked by their expression levels. How could such an approach be able to achieve the performance as shown in Figure 1F? Are CIGs a subset of highly expressed genes? Equally interesting is that a single histone feature can also lead to similar prediction accuracy as RNA expression. I found this hard to believe. What is the underpinning biology supporting this?

A: This is a very good question. Indeed, CIGs have a relatively high expression, but not all the highly expressed genes are CIGs.

(1) Highly expressed genes are not enriched in cell type related pathways, whereas CIGs are enriched in these pathways (**Fig 1G-H**).

(2) As shown in **Fig S3C**, RNA expression rank performed very well in recapturing the negative control genes as indicated by the blue curve, but performed very poor in recapturing cell identity genes indicated by the red curve. This is because of the relatively low expression level of the negative control genes. When we require the negative control genes to have expression levels similar to that of the positive CIGs (**Fig R2A**), CIGdiscover still have considerably good performance (**Fig R2B**), but the RNA expression rank fails to distinguish between these negative control genes and the positive CIGs.

(3) The underpinning biology supporting the performance histone feature in predicting cell identity genes is as we have summarized above: cell identity genes are different from other expressed genes in the epigenetic mechanism to activate their transcription. Epigenetic markers not only indicate activation or repression of a gene, but also, further indicate different ways in which different genes are activated or repressed. For example, although both housekeeping genes and cell identity genes are all activated, housekeeping genes show sharp enrichment of H3K4me3 in promoter, whereas cell identity genes show broad H3K4me3 that covers both promoter and gene body^{4,9}. Similarly, cell identity genes tend to be regulated by super enhancers, whereas many other expressed genes are regulated by typical enhancers¹⁰⁻¹². Therefore, compare to expression level, a combination of epigenetic signatures can be better indicator of cell identity genes.

3. Given the above observation, the definition of negative control genes does not appear to properly control for CIGs. Specifically, for a given cell type, the authors randomly selected from genes that do not appear with that cell type in literature and other major databases (page 11; line 481-486). Control genes selected this way will most likely to display much lower expression levels control to CIGs because CIGs are by definition expressed in the cell type of consideration. Given the high prediction power shown in Figure 1F from using RNA expression alone, the authors should at least select control genes that have similar levels of expression to CIGs.

A: As also described above, we randomly selected a group of genes but require similar expression level as that of the positive CIGs (**Fig R2A**) and used the same approach to build CIGdiscover. As shown in **Fig R2B**, compared to the model only built by RNA expression, CIGdiscover built by epigenetic signature still have a much better performance. However, considering that most non-identity genes in a cell would have low expression, we still choose to not exclude the low expression genes from the pool of negative controls, so that the model could be closer to the reality in the cell.

4. The number of CIGs curated for each cell type is relatively small (Figure 1A and B). Clearly, there are a large number of CIGs that are not curated for each cell type and they are what the authors attempt to predict for. Then, how would the authors be sure that when they randomly select for control genes for CIGs, these randomly selected genes themselves are not CIGs? The author should take this into account when selecting for control genes, so the learning model will not over penalize potential CIGs because some of the unknown CIGs were accidentally selected as control genes.

A: This is a very good question. To reduce the possibility of including potential CIGs as control genes and thus over penalizing potential CIGs, we have removed genes that appear with that cell type in literature and in other major databases. To investigate potential effect of false control genes, we performed robustness test for CIGdiscover. The results indicated that our model is quite resilient to false positive or false negative training genes. Even when 30% of the control genes were from CIGs and thus are false control genes, the AUC of ROC only reduced from 0.93 to 0.89 (**Fig S11C**). As expected, the performance of CIGdiscover was ruined when the noise ratio reached 50%, suggesting that the good performance at low noise ratio was not due to an over fitting effect.

5. A grid-based genetic method (GridGO) was used for finding the optimal ChIP-seq peak height, boundary, and several other characteristics that can improve the prediction accuracy of CIG. This procedure does not appear to be nested in any cross-validation procedure. As such, the optimization of ChIP-seq parameters will lead to overfitting

because the histone features are optimized for CIG prediction using training data prior to the feature selection and prediction steps.

A: we performed the cross validation for GridGO optimization to test whether this approach caused over fitting (**Fig R3A**). We first split the datasets and used 80% of genes to perform GridGO optimization and train the CIGdiscover model. Afterwards, we tested the model performance using the 20% remaining genes, which were not used in GridGo optimization and the training of CIGdiscover. The results based on 100 times cross validation indicated that the GridGO optimization did not cause over-fitting (**Fig R3B**). This result also agrees with our observation that the CIGdiscover trained by 1 subcategory of CIGs can predict the other 3 subcategories of CIGs (**Fig 2E, S11D**).

6. The feature selection step appears to be nested in a random split of training and test procedure (as depicted in Figure S3A). However, it is not clear why the 9 features (Figure 1D) were always selected from the forward feature selection procedure. Due to the random split of the training and test data, different number and different combination of features are expected to be found.

A: To better illustrate the procedure, we have revised the **Fig S3A** to show the iterative feature selection process. In an iteration i , let $S_{i-1}=[s_1, s_2, \dots, s_{i-1}]$ be the combination of features selected by the previous $i-1$ iterations, and let $C_{i-1}=[c_1, c_{i+1}, \dots, c_n]$ be the remaining candidate features. Our algorithm will combine c_i with S_{i-1} to form a new candidate combination, and evaluate the performance of this combination by 100 times cross validations. Similarly, the algorithm will combine $c_{i+1}, c_{i+2}, \dots, c_n$ with S_{i-1} to form $n-i$ additional candidate combinations, and evaluate the performance of each candidate combination by 100 times cross validations. Among these $n-i+1$ candidate combinations, the combination that shows the best performance will be the combination S_i selected by iteration i . Due to the large number of cross validation at each iteration, the 9 features were consistently selected.

7. The authors found that the combination of H3K4me1/3 and H3K27ac can accurately identify CIGs within and across multiple cell types (Figure S5A) and across the four categories of CIGs (Figure S5D). This implies that the same combination of H3K4me1/3 and H3K27ac features define CIGs in all cell types considered in this study. It is known that the presence of H3K4me1/3 and H3K27ac marks are associated with active genes. How could such a combination be specific to CIGs and at the same time be non-cell-type-specific is surprising and at the same time hard to understand. Could this be an artifact caused by overfitting per my comments in (5) and (6)?

A: (1) It is true that CIGs and some other active genes can both be marked by an activating histone modification, e.g., H3K4me3. However, the pattern of this histone modification at CIGs can be different from the pattern of this histone modification at other active genes, e.g., CIGs show broad enrichment of H3K4me3, whereas housekeeping genes show sharp enrichment peak of H3K4me3. Therefore, by analyzing different patterns (signatures) of a histone modification, we are able to distinguish between CIGs and other active genes in the same cell type. (1) Given two cell types A and B, CIGs of cell type A are different from CIGs of cell type B. The rationale for CIGdiscover is that the pattern of a histone modification at CIGs of Cell type A is similar to the pattern of this histone modification at CIGs of cell type B. For example, CIGs of cell type A and CIGs of cell type B both show broad pattern of H3K4me3. Therefore, using histone modification data from cell type A and CIGs of cell type A to train our model, we will be able to know that a combination of histone modification patterns x, y, z are associated with CIGs. Thereafter, we will use histone modification data from cell type B to look for genes that

are marked by the features x, y, and z, and predict these genes to be CIGs of cell type B.

8. There are multiple ways to calculate a specificity score (PMC5444245) from multiple cell types given the RNA expression data. Would the specificity score calculated from using alternative methods (alternative to the “Tau” method used in (Figure S7B)) give equal or better performance in identifying CIGs than such a machine learning approach? The authors should perform a “cross-validation” by using different specificity scores to quantify the performance and compare their machine learning predicted genes with genes found by an alternative specificity metric to the one that is used for quantifying the performance.

A: Following our reviewer’s suggestion, we used all different methods described in the publication PMC5444245 to calculate specificity score, and compared the performance between these scores and CIGdiscover in identifying CIGs. CIGdiscover significantly outperformed these specificity scores (**Fig R4A**). In addition, enrichment in EC pathways is much more significant for CIGs predicted by CIGdiscover when compared to genes predicted by individual specificity scores (**Fig R4B**). Notably, Different from specificity scores, which rely on expression values from many biosamples, CIGdiscover only requires the epigenetic information from the target cell type to achieve reliable prediction. Together, we conclude that CIGdiscover built with epigenetic signatures are a better than expression specificity scores in identifying CIGs.

Response Figure Legends

Fig. R1

Figure R1. Endothelial CIGs showed little effect on the proliferation of Normal Human Lung Fibroblasts (NHLF). (A) UCSC Genome Browser tracks to show the expression of individual genes in NHLFs. (B) Proliferation rate of NHLF with or without individual genes knocked out by CRISPR-Cas9 system. T7 Endonuclease cleavage assay was used to confirm cutting efficiency of the CRISPR-Cas9 system, as indicated at the bottom of figures for individual genes. P values determined by Student's T test. *, $P < 0.05$.

Fig. R2

Figure R2. CIGdiscover still performed well after requiring the negative control genes and positive CIGs in the training set to have similar expression level. (A) Boxplot to show expression values of the positive CIGs and negative control genes. (B) ROC curves to show performance of CIGdiscover and RNA expression rank in predicting the positive CIGs and negative control genes after requiring these two gene sets to have similar expression level.

Fig. R3

Figure R3. Cross-validation to show that optimization of histone modification features for CIGs by GridGO did not cause an over fitting effect. (A) Flowchart of the cross-validation. (B) ROC curves to show the performance of CIGdiscover.

Fig. R4

Figure R4. CIGdiscover performed better than expression specificity scores in predicting CIGs. (A) ROC curves to show the performance of CIGdiscover and individual expression specificity scores. (B) Enrichment of EC pathways in CIGs predicted by CIGdiscover or individual expression specificity scores.

Response References

- 1 Liu, C. R., Berry, P. M., Dawson, T. P. & Pearson, R. G. Selecting thresholds of occurrence in the prediction of species distributions. *Ecography* **28**, 385-393 (2005).
- 2 Hanley, J. A. & McNeil, B. J. A method of comparing the areas under receiver operating characteristic curves derived from the same cases. *Radiology* **148**, 839-843, doi:10.1148/radiology.148.3.6878708 (1983).
- 3 Hanley, J. A. & McNeil, B. J. The meaning and use of the area under a receiver operating characteristic (ROC) curve. *Radiology* **143**, 29-36, doi:10.1148/radiology.143.1.7063747 (1982).
- 4 Chen, K. *et al.* Broad H3K4me3 is associated with increased transcription elongation and enhancer activity at tumor-suppressor genes. *Nat Genet* **47**, 1149-1157, doi:10.1038/ng.3385 (2015).
- 5 O'Connor, T., Boden, M. & Bailey, T. L. CisMapper: predicting regulatory interactions from transcription factor ChIP-seq data. *Nucleic Acids Res* **45**, e19, doi:10.1093/nar/gkw956 (2017).
- 6 Buckle, A., Brackley, C. A., Boyle, S., Marenduzzo, D. & Gilbert, N. Polymer Simulations of Heteromorphic Chromatin Predict the 3D Folding of Complex Genomic Loci. *Mol Cell* **72**, 786-797 e711, doi:10.1016/j.molcel.2018.09.016 (2018).
- 7 Singh, R., Lanchantin, J., Robins, G. & Qi, Y. DeepChrome: deep-learning for predicting gene expression from histone modifications. *Bioinformatics* **32**, i639-i648, doi:10.1093/bioinformatics/btw427 (2016).
- 8 Roy, S. *et al.* A predictive modeling approach for cell line-specific long-range regulatory interactions. *Nucleic Acids Res* **43**, 8694-8712, doi:10.1093/nar/gkv865 (2015).
- 9 Benayoun, B. A. *et al.* H3K4me3 breadth is linked to cell identity and transcriptional consistency. *Cell* **158**, 673-688, doi:10.1016/j.cell.2014.06.027 (2014).
- 10 Whyte, W. A. *et al.* Master transcription factors and mediator establish super-enhancers at key cell identity genes. *Cell* **153**, 307-319, doi:10.1016/j.cell.2013.03.035 (2013).
- 11 Feng, B. *et al.* Reprogramming of fibroblasts into induced pluripotent stem cells with orphan nuclear receptor Esrrb. *Nat Cell Biol* **11**, 197-203, doi:10.1038/ncb1827 (2009).
- 12 Hnisz, D. *et al.* Super-enhancers in the control of cell identity and disease. *Cell* **155**, 934-947, doi:10.1016/j.cell.2013.09.053 (2013).

Reviewers' comments:

Reviewer #1 (Remarks to the Author):

The authors have addressed all my comments. There are just a couple of minor issues that need to be fixed:

1. The summary table in Fig. 1J has a couple of mistakes in it (or some of the labels in Fig. S10 are incorrect). The "X" in EGFL7 EC branching should be a check (i.e., it significantly reduced branching) and the check in PTGER4 EC branching should be an "X" (i.e., it did not significantly reduce branching) based on the data/results shown in Fig. S10. Additionally, this panel in Fig. 1 is mislabeled as "H". It should be labeled "J".

2. In Fig. S10A & B, "PTGER4" is mislabeled as "PTER4" (i.e., "G" was dropped).

3. In the training CIGdiscover section, the loglikelihood takes the natural form for logistic regression (i.e., all the responses in the sample are independently Bernoulli distributed). But it seems like the natural approach for applying an L1 penalty is to minimize something like $-L(\theta) + \lambda \sum_{j=1}^p |\beta_j|$ with respect to the parameters θ/β .

Instead, the authors go on to describe lasso regression associated with L1 penalty of a linear regression model? $L(\theta)$ and $g(\theta)$ are not clearly related to each other, $g(\theta)$ has nothing to do with logistic regression and it's not clear how the logistic regression itself is L1 penalized.

4. In the Backward and forward feature selections, the "c_I" (with the capital "I") can be replaced with "c_i" (lower case "I").

5. Line 276, "potential be" should be "potential to be".

6. Line 615, "ration" should be "ratio".

Best,
Stefan

Reviewer #2 (Remarks to the Author):

The authors have addressed to each of my comments. I have the following remaining concerns:

The manuscript lacks a clear definition of "cell identity gene". The authors' responded that "although some cell identity genes can be very specific to one cell type, some other cell identity genes might be expressed in related cell types and have different degree of expression specificity...". While I agree with the authors that cell identity genes may be expressed in closely related cell types, they cannot be expressed in a wide range of cell types (such as housekeeping genes) if they are to establish specific cell types. Are there any references supporting the conceptual differences between "cell identity genes" and cell type-specific genes in terms of specifying cell types? I have not found any publication nor did the authors cite any references that define and distinguish these types of genes.

I still believe there is a strong overfitting in CIG prediction according to Figure 1F. Simply by ranking the genes based on their expression levels, the authors were able to predict more than 50% of CIGs at very low false positive rate. This may be resolved by building much larger positive training sets. At the moment, the largest positive training set only contains 36 CIGs while the smallest one contains 18 CIGs. Can such small datasets be reliably used for benchmarking performance in CIG identification?

The authors appear to combine CIGs from all 10 cell types. Why assuming they have the same properties in different cell types? To this end, the authors suggested that CIGs have broad H3K4me1/3 and H3K27ac. If the authors take CIGs from their approach and compare to "CIGs" predicted using RNA expression level only (i.e. black line, Figure 1F), would there be any difference as to the width of the above marks among the two groups?

Summary: We thank both reviewers for reading our manuscript again very carefully, for acknowledging that we have addressed each of the previous comments, and for further providing the additional constructive comments. In the new manuscript, we have performed new analyses and revised our writing as suggested by each reviewer, indicated by Figures S1A, S1D, S5-8, S15B, S16 I-J, R1, and the texts highlighted in yellow color in the new manuscript. We are confident that the revised manuscript has been substantially improved to fit the high quality of manuscript for *Nature Communications*.

Reviewers' comments:

Reviewer #1 (Remarks to the Author):

The authors have addressed all my comments. There are just a couple of minor issues that need to be fixed:

1. The summary table in Fig. 1J has a couple of mistakes in it (or some of the labels in Fig. S10 are incorrect). The "X" in EGFL7 EC branching should be a check (i.e., it significantly reduced branching) and the check in PTGER4 EC branching should be an "X" (i.e., it did not significantly reduce branching) based on the data/results shown in Fig. S10. Additionally, this panel in Fig. 1 is mislabel as "H". It should be labeled "J".

Response: Thank you! We feel sorry for the mislabels in Fig. 1J. The "X" in EGFL7 EC branching has been changed to be a check, and the check in PTGER4 EC branching has been changed to be an "X". This panel has been labeled as "J" now.

2. In Fig. S10A & B, "PTGER4" is mislabeled as "PTER4" (i.e., "G" was dropped).

Response: Thank you! We feel sorry for the typo. We have changed it to "PTGER4", and double-checked the whole manuscript to ensure that this typo does not exist now.

3. In the training CIGdiscover section, the loglikelihood takes the natural form for logistic regression (i.e., all the responses in the sample are independently Bernoulli distributed). But it seems like the natural approach for applying an L1 penalty is to minimize something like $-L(\theta) + \lambda \sum_{j=1}^p |\beta_j|$ with respect to the parameters θ/β . Instead, the authors go onto describe lasso regression associated with L1 penalty of a linear regression model? $L(\theta)$ and $g(\theta)$ are not clearly related to each other, $g(\theta)$ has nothing to do with logistic regression and it's not clear how the logistic regression itself is L1 penalized.

Response: We thank Dr. Bekiranov for finding this problem! Because we used the Sklearn logistic regression library, so the problem is in the writing of this manuscript and did not happen in our software. We addressed the issue by citing the correct equation for logistic regression, and added the following details regarding our implementation of logistic regression in the Online Methods section (**highlighted by yellow color between lines 643-650 in the manuscript**):

The Lasso method is implemented by fixing an upper bound on the sum of the absolute value of the model parameters, which can be denoted by penalizing the negative log-likelihood with L1-norm. In the Logistic regression model, the negative log-likelihood is denoted by

$$-\sum_{i=1}^n \log(p_{\beta}(y_i|x_i)) = \sum_{i=1}^n \left\{ -y_i \left(\sum_{j=0}^p \beta_j x^{(j)} \right) + \log \left(1 + \exp \left(\sum_{j=0}^p \beta_j x^{(j)} \right) \right) \right\}.$$

The loss function ρ can be written as

$$\rho_{(\beta)}(x, y) = -y\left(\sum_{j=0}^p \beta_j x^{(j)}\right) + \log\left(1 + \exp\left(\sum_{j=0}^p \beta_j x^{(j)}\right)\right).$$

The Lasso estimator of a Logistic regression model is defined as

$$\hat{\beta}(\lambda) = \operatorname{argmin}_{\beta} \left(n^{-1} \sum_{i=1}^n \rho_{(\beta)}(x_i, y_i) + \lambda \|\beta\|_1 \right).$$

4. In the Backward and forward feature selections, the "c_I" (with the capital "I") can be replaced with "c_i" (lower case "I").

Response: Thank you for this suggestion! We have changed the "I" to "i" in the Methods section.

5. Line 276, "potential be" should be "potential to be".

Response: Thank you! We have changed it to "potential to be". (Line 298 in the manuscript)

6. Line 615, "ration" should be "ratio".

Response: Thank you! We have changed it to "ratio". (Line 655 in the manuscript)

Reviewer #2 (Remarks to the Author):

The authors have addressed to each of my comments. I have the following remaining concerns:

The manuscript lacks a clear definition of "cell identity gene". The authors' responded that "although some cell identity genes can be very specific to one cell type, some other cell identity genes might be expressed in related cell types and have different degree of expression specificity...". While I agree with the authors that cell identity genes may be expressed in closely related cell types, they cannot be expressed in a wide range of cell types (such as housekeeping genes) if they are to establish specific cell types. Are there any references supporting the conceptual differences between "cell identity genes" and cell type-specific genes in terms of specifying cell types? I have not found any publication nor did the authors cite any references that define and distinguish these types of genes.

Response: This is a very good question. We are sorry that we failed to explain it clear enough in the last version of our manuscript and thus, have further revised the related part in the new manuscript (highlighted by yellow color at lines 63-75, 90-97, 103-105, 164-166 and 218-224 in the manuscript).

For the following three reasons, we decided to directly predict cell identity genes based on epigenetic signatures and not to predict cell identity genes based on expression specificity. (1) As agreed by our reviewer, cell identity genes may be expressed in closely related cell types, but cannot be expressed in a wide range of cell types. Therefore, the relationship between cell identity genes and cell type specific genes can be complicated and uncertain: some cell identity genes might be highly cell type specific, whereas other cell identity genes can be less specific. Our analysis indicated that the cell identity genes reported in literature have a high but a widely distributed Tau index of expression specificity, with their index values ranging from approximately 0.6 to

1 and ranked from top to median among all genes (**Fig. S1D**). Therefore, although our CIGdiscover algorithm didn't utilize information related to cell type specificity, the CIGs predicted by CIGdiscover tend to show an **overall** strong specificity to their associated cell types (**Fig. S4D**), but these **individual CIGs** did show a wide range of specificity (**Fig. 4B, S17D**). (2) Further, expression specificity analysis requires comparison between a query cell type and most, if not all, other cell types. It also requires distinguishing between cell-type-specific and biological-condition-specific genes, e.g., genes induced by a cell culture condition. Therefore, it is not cost-effective, if not impossible, to collect data for all cell types under all biological conditions for the expression specificity analysis. Therefore, a published method¹ that utilized expression specificity to define cell identity genes appeared to have the worst performance when compared to our approach and other conventional methods such as network analysis (**Fig. 3E**). (3) It becomes apparent recently that CIGs are different from other expressed genes in the epigenetic mechanism to regulate their transcription. For examples, super enhancers and a unique broad pattern of H3K4me3 modification were found to regulate CIGs, whereas it is typical enhancer and sharp H3K4me3 modification that regulate other expressed genes such as housekeeping genes²⁻⁵. Notably, analyzing these signatures only need comparison between different genes in a single investigated cell type and does not need comparison between cell types.

It is true that although thousands of literature directly used the word “cell identity gene”, to our knowledge, no reference clearly defined the concept of cell identity gene yet. There is also no reference reporting difference between cell identity genes and cell type-specific genes, probably just because the relationship between these two concepts can be complicated (as described above). Another reason might be that “cell identity gene” is a new concept that was increasingly used in recent years. A Google Scholar search returned 2,960 publications that used the phrase “cell identity genes”, whereas there were only 187 and 20 publications that used this phrase before 2009 and before 1999, respectively (**Figure R1**). Therefore, we are proud to be the first to provide a clear definition of “cell identity gene” based on our thorough review of literature.

Specifically, our thorough review of literature that used the phrase “cell identity gene” motivated us to define cell identity genes as belonging to one of four functional categories: (1) master transcription factors, which drive the differentiation towards a cell type when their expression is ectopically induced in another cell type; (2) required transcription factors, whose depletion impaired the differentiation towards a specific cell type, (3) genes required for key functions or phenotypes of a cell type; (4) genes that were widely used as markers for a cell type. Notably, our CEF-CIG model trained by one of these four category can predict the other three categories, with an accuracy similar to the model trained by all four category together (**Figure S15D**). Therefore, the result from this test indicated that the four functional category of cell identity genes actually have similar epigenetic signatures.

We have further provided a publicly accessible database of manually curated cell identity genes reported in literature, as well as our computationally predicted identity genes for individual cell or tissue types. We hope that these works will make it easier for many researchers to study cell identity genes in future.

I still believe there is a strong overfitting in CIG prediction according to Figure 1F. Simply by ranking the genes based on their expression levels, the authors were able to predict more than 50% of CIGs at very low false positive rate. This may be resolved by building much larger positive training sets. At the moment, the largest positive training set only contains 36 CIGs while the smallest one contains 18 CIGs. Can such small datasets be reliably used for benchmarking performance in CIG identification? The authors appear to combined CIGs from all 10 cell types. Why assuming they have the same properties in different cell types?

Response: We thank the reviewer for these very constructive questions. There are three concerns in this comment: (1) on the good performance of the model that were built only by the RNA expression level, (3) on combining CIGs from multiple cell types, and (2) on the size of training dataset. During the development of our models, we also had the same concerns. We feel sorry that we did not explain these clear enough in the previous version and thus, have revised the manuscript accordingly (associated lines in the manuscript are indicated below):

(1) Concern on the good performance of the model based on gene expression level: In addition to ROC curve analysis, we further performed several additional types of analyses to comprehensively test and understand different aspects of the performance of these alternative models for CIG prediction. For instance, these analyses indicated that RNA expression level is good at distinguishing negative control genes from other genes, but is not good at distinguishing positive cell identity genes from other genes (**Figure S3C bottom-middle panel**). This explained why the false positive rate for the model that was based only on RNA expression level could still be low when the true positive rate reaches to a medium level of 50% (**Figure 1F**). Further, the analyses showed that cell identity genes predicted by epigenetic signatures are enriched in cell type related pathways, but high expression genes (genes predicted by model only with RNA expression level) are not (**Fig 1G-H, Fig 4D**). Considering that we have done 100 times cross validation, together with the additional performance analyses described here, we hope that we have convinced our reviewer that although RNA expression level can predict cell identity genes to certain degree of accuracy, this prediction is overall not optimal and is not an overfitting (Related context has been highlighted by yellow color at Lines 138-141 and 148-150 in the manuscript)

(3) Concern on combining CIGs from all 10 cell types: First, knowledges in literature and results from our analysis both suggested that it is reasonable to combined genes from multiple cell types to gain a strong statistical power. Both literature and our analysis indicated that the existence of unique epigenetic signatures (e.g., broad H3K4me3 and super enhancer) at cell identity genes but not at other genes (e.g., house keeping genes) is a common phenomenon across different cell types^{3,5}. For instance, embryonic stem cell (ESC) identity genes displayed broad H3K4me3 in ESCs, and neural cell identity genes also displayed broad H3K4me3 in neural cells³. We also verified that the manually curated cell identity genes each showed broad H3K4me3, H3K4me1, and H3K27ac in their associated cell types (**Fig S1A-C**). Therefore, when our machine-learning model learned that stem cell identity genes display features such as broad H3K4me3 in ESCs, it would be able to use the broad H3K4m3 in neural cells to define identity genes of neural cells. Also, when we combine CIGs of ESCs and their H3K4me3 signatures in ESC with CIGs of neural cells and their H3K4me3 signatures in neural cells, the machine-learning model will be able to learn that CIGs of ESCs and CIGs of NPCs both display broad

H3K4me3 in their associated cell types. Therefore, the model will be able to use broad H4K4me3 in a third cell type to predict cell identity genes for that third cell type. Second, our multiple tests further verified that it is good to combine reported CIGs from multiple cell types to train the model. As our reviewer has noted, the number of reported cell identity genes for each cell type is small. Therefore, our final training datasets combined 247 manually curated cell identity genes from 10 different cell types. Our test indicated that combining CIGs from multiple cell types did improved the prediction, although up to 3 cell types will be good enough (**Fig 2C-D**). The test further verified that the prediction accuracy is similar between a model trained by data from a set of cells to predict CIGs for this same set of cells (parallel test) and a model trained by data from a set of cells to predict CIGs for a different set of cells (cross test) (**Fig S15A**). (Related context has been highlighted by yellow color at Lines 98-102 and 178-196 in manuscript)

(3) *Concern on the size of training dataset:* In response to our reviewer's suggestion, we have tested the models by increasing the size of training dataset. The good performance of the model that is solely based on gene expression level is due to its ability to recapture negative control genes but not the positive CIGs (**Figure S3C bottom-middle panel**). However, our result indicated that further increasing the number of negative control genes has little influence on the performance of the full model or the model based on only RNA expression level (**Figure S15B**). Because the number of comprehensively validated CIGs in literature is small, it is hard to further manually curate additional positive CIGs for a cell type that also has the epigenomic data available. However, down-sampling analysis indicated that the number of positive CIGs has also been large enough to saturate the performance. The analysis indicated that further increasing the number of positive CIGs will also have little effect on the performance of the full model (**Figure 2C-D**) or the model based on only RNA expression level (**Fig S16I-J**). Therefore, building a larger training dataset and down sampling the training dataset both indicated small effect of the current size of training data on the performance of these models. (Related context has been highlighted by yellow color at Lines 199-201 and 205-209 in the manuscript)

To this end, the authors suggested that CIGs have broad H3K4me1/3 and H3K27ac. If the authors take CIGs from their approach and compare to "CIGs" predicted using RNA expression level only (i.e. black line, Figure 1F), would there be any difference as to the width of the above marks among the two groups?

Response: We Thank the reviewer for this constructive question and suggestion! We performed the analysis accordingly. Genes predicted using RNA expression level showed significantly higher expression level when compared to random control genes or CIGs predicted by the full model, whereas CIGs predicted by our approach are marked by the broader H3K4me1/3 and H3K27ac when compared to random control genes or genes predicted by RNA expression (**Fig S5-8**). Therefore, although RNA expression can predict some cell identity genes, the prediction is still not as good as that of our approach and tend to capture high expression genes rather than genes marked with the reported epigenetic signatures of cell identity genes. We also performed several additional analyses to compare the putative CIGs predicted by our approach (the full model) with the genes predicted using RNA expression level only. As shown in **Fig 1G, 1H and 4D**, genes predicted using only RNA expression level failed to significantly enrich in the cell type-related functional pathways. (Related context has been highlighted by yellow color at Lines 146-150 and 157-163 in manuscript).

Response References:

1. Cinghu S, Yellaboina S, Freudenberg JM, Ghosh S, Zheng XF, Oldfield AJ, Lackford BL, Zaykin DV, Hu G and Jothi R. Integrative framework for identification of key cell identity genes uncovers determinants of ES cell identity and homeostasis. *P Natl Acad Sci USA*. 2014;111:E1581-E1590.
2. Chen K, Chen Z, Wu D, Zhang L, Lin X, Su J, Rodriguez B, Xi Y, Xia Z, Chen X, Shi X, Wang Q and Li W. Broad H3K4me3 is associated with increased transcription elongation and enhancer activity at tumor-suppressor genes. *Nat Genet*. 2015.
3. Benayoun BA, Pollina EA, Ucar D, Mahmoudi S, Karra K, Wong ED, Devarajan K, Daugherty AC, Kundaje AB, Mancini E, Hitz BC, Gupta R, Rando TA, Baker JC, Snyder MP, Cherry JM and Brunet A. H3K4me3 breadth is linked to cell identity and transcriptional consistency. *Cell*. 2014;158:673-88.
4. Hnisz D, Abraham BJ, Lee TI, Lau A, Saint-Andre V, Sigova AA, Hoke HA and Young RA. Super-enhancers in the control of cell identity and disease. *Cell*. 2013;155:934-47.
5. Whyte WA, Orlando DA, Hnisz D, Abraham BJ, Lin CY, Kagey MH, Rahl PB, Lee TI and Young RA. Master transcription factors and mediator establish super-enhancers at key cell identity genes. *Cell*. 2013;153:307-19.

REVIEWERS' COMMENTS:

Reviewer #1 (Remarks to the Author):

The authors have addressed all my remaining minor issues.

Best,
Stefan Bekiranov

Reviewer #2 (Remarks to the Author):

The authors have responded comprehensively to the concerns I've raised. While most of the responses are satisfactory, regarding the prediction based on gene expression, the authors suggested that gene expression is only useful at distinguishing negative control genes but not good at capturing positive CIGs (Figure S3C bottom-middle panel). I note that results from Figure S3C appear to be inconsistent with Figure 1F where prediction from using gene expression has comparable false positive rate (that distinguishes negative control genes) and true positive rate (that captures positive CIGs) to those obtained from using the width of different histone marks.

Summary: We thank all reviewers for reading our manuscript very carefully, for acknowledging the novelty and significance of our work, and for providing many constructive comments. We are glad that both reviewers acknowledged that we have addressed all their previous concerns. We are optimistic that the new question of the reviewer #2 has also been addressed satisfactorily in the new response letter enclosed here.

Reviewer #2 (Remarks to the Author):

The authors have responded comprehensively to the concerns I've raised. While most of the responses are satisfactory, regarding the prediction based on gene expression, the authors suggested that gene expression is only useful at distinguishing negative control genes but not good at capturing positive CIGs (Figure S3C bottom-middle panel). I note that results from Figure S3C appear to be inconsistent with Figure 1F where prediction from using gene expression has comparable false positive rate (that distinguishes negative control genes) and true positive rate (that captures positive CIGs) to those obtained from using the width of different histone marks.

Response: We feel sorry for the new confusion triggered by our response to the previous concern. Since almost all the control genes are assigned with very low CIG probability scores by the RNA expression model (**Fig S3C**), it could achieve a good true positive rate by using a low probability cutoff but still maintain a low false positive rate. This behavior is hard to observe in ROC curve (**Fig 1F**), which is a mixed result of the true positive rate and false positive rate. This is why it is important to always further perform cumulative distribution analyses of true and false positive rate separately (and we feel proud that we have done so), although such situation does not happen frequently and thus researchers often do not further show such figures when it did not happen. Indeed, our cumulative distribution analysis and other additional analyses (**Figure 1G-H, 4D**) consistently indicated the poorer overall performance of the RNA expression model when compared to our CIGdiscover. We hope our concise explanation will have satisfactorily answered this good question of the reviewer.